# Bandit Learning for Online Scheduling with Immediate Decision

## Abstract

Online scheduling has been extensively studied in computer science and economics owing to its broad applications. Motivated by streaming task processing in domains such as IoT data streaming and cloud resource allocation, we investigate an online scheduling setting where the scheduler must immediately decide whether to accept an incoming task. Consider a system with $M$ identical machines. At each time step, multiple tasks arrive, and each machine must immediately assign itself to a task or remain idle. Tasks that are not processed immediately are abandoned and cannot be revisited. Upon completion, a task yields a reward, which may be stochastic and initially unknown. Through repeated task completions, the scheduler can learn the reward distributions over time. In this work, we formalize this problem as online scheduling with immediate decision. We first analyze the setting with known rewards, for which we derive a worst-case competitive ratio and propose a near-optimal online algorithm. For the case of unknown and random rewards, we design an efficient bandit algorithm that balances exploration and exploitation, achieving an $O(\log T)$ regret over a time horizon $T$. Experimental results demonstrate the efficacy of the proposed algorithms.

## 1 Introduction

Online scheduling has long been at the forefront of research in the fields of operations research, computer science, and management science (Graham, 1966; Pruhs et al., 2004; Wang et al., 2024). Its prevalence can be attributed to the wide array of real-world applications like cloud computing, crowdsourcing and multiprocessing systems. This concept deals with the dynamic allocation of resources to tasks as they arrive over time, without knowing the prior knowledge about future tasks. In this paper, our goal is to maximize the sum of rewards for completed tasks, where the reward for each task represents its value and significance. For example, the task in the cloud computing system has a reward determined by its type, significance, necessity, and some other properties. Tasks that require a large amount of computing resources and are urgent usually have higher rewards, and the cloud computing system endeavors to complete as many high-reward tasks as possible.

In various streaming task scheduling applications such as IoT data processing (Hou et al., 2009), cloud computing resource scheduling (Zhou, 2024; Wang et al., 2024), and financial data processing, the coming tasks need to be processed immediately. For example, in IoT data processing, numerous sensors continuously generate data streams (Wang et al., 2022b). These data packets arrive at the processing unit in real-time, and immediate decisions are required on which data to process first. Tasks that are not processed immediately will be ignored and will no longer have the chance to be processed (Wang et al., 2022a). If the system fails to prioritize correctly, it could lead to unforeseen equipment breakdowns and production delays (Serrano-Ruiz et al., 2021). Considering the unique nature of this immediate decision-making requirement in these streaming task applications, which has been overlooked in previous research, this paper models and analyzes the online scheduling with immediate decision-making problem.

Moreover, in the context of our online scheduling problem inspired by applications like IoT and financial data processing, the reward associated with each task poses a significant challenge due to its uncertainty and randomness (Cayci et al., 2019; Xu et al., 2024). In IoT data processing, for instance, "reward" obtained from processing a particular data stream is contingent upon multiple factors. The nature of the data itself, such as whether it contains critical information for system opti-

mization or is merely supplementary, can impact the reward. Similarly, in financial data processing, the gain from processing a transaction data task can be highly unpredictable. Market fluctuations, the type of transaction, and the involved parties can all contribute to the randomness of the reward. However, these uncertainties can usually be learned through multiple task completions since it reflects on some invariant properties of that task. In this paper, we model the randomness of rewards by introducing the notion of "type" for each task. Each type is associated with a fixed reward distribution, which is unknown to the learner but can be learned through multiple completions. For example, in cloud computing system, tasks can be divided into different types by their goals, requirements and outcomes (Xu et al., 2024). Multi-armed bandit (MAB) serves as a crucial tool for making decisions under uncertainty in multiple rounds (Lattimore & Szepesvári, 2020). This tool typically involves iteratively selecting multiple arms, with each arm possessing a distinct reward distribution unknown to the player. The player's objective is to minimize their cumulative regret, defined as the expectation of the difference in cumulative rewards between the arm with the highest reward and the player's selected arm over $T$ rounds. Since only the reward of the pulled arm is observed, the player must balance between exploration (acquiring information on the different arms) and exploitation (pulling the seemingly best arm). Leveraging the powerful decision-making capabilities of the MAB tool, we propose a novel algorithm to address the reward learning problem in our online scheduling setting.

To address the problem of the online scheduling problem with immediate decision-making and uncertain rewards, this paper adopts a systematic approach. Our contributions are summarized as follows.

- In Section 3, we formulate a comprehensive model for this variant of online scheduling and justifies its importance. There are $M$ identical machines that can process tasks. At each time a set of tasks arrive, and the machine must decide immediately which tasks to process. Each task has a fixed processing length and a type, which is associated with an unknown fixed reward distribution. The machine will receive the reward once it completes the task. Preemption is allowed but the preempted task will never be revisited again. The objective for a scheduling algorithm is to maximizes the total rewards of completed tasks.

- We first consider the case of realizing reward when the task arrives in Section 4, which is a classic setting in the literature of online scheduling. We establish the worst-case competitive ratio bound of $\tilde{O}\left(1/(ML_{\max}^{1/M})\right)$ [1], where $L_{\max}$ is the maximum processing length. The competitive ratio is defined as the obtained reward ratio between an online algorithm and the optimal offline algorithm. In addition to this, a scheduling algorithm is proposed tailored to this known-reward case. The "maximum remaining density first" (MRDF) algorithm attains at least $\Omega\left(1/(ML_{\max}^{1/M})\right)$-competitive ratio, indicating that it is near-optimal compared with the worst-case.

- In Section 5, we turn our attention to the more realistic and challenging scenario where the reward for each task is unknown and random. An efficient bandit-based algorithm named "scheduling upper confidence bound (S-UCB)" is introduced, which uses UCB index to replace reward for each task's type and runs the MRDF algorithm. The algorithm is proven to attain an $O(\log T/\Delta)$ sublinear approximate regret, where $T$ represents the horizon length and $\Delta$ is the minimum density gap. The regret is defined as the difference of total rewards between the learning algorithm and the MRDF algorithm given knowledge of reward. This result demonstrates that the algorithm's performance converges to the MRDF algorithm. We also provide the competitive ratio analysis for this algorithm, showing that it asymptotically converges to $\Omega\left(1/(ML_{\max}^{1/M})\right)$ as $T$ increases. Additional experiments also show the efficiency and convergence of this online bandit-based algorithm.

## 2 RELATED WORK

**Classic online scheduling.** Online scheduling is a fundamental area of research in operations research and computer science, focusing on algorithms that make decisions based on partial information, with tasks arriving over time. The investigated problem settings, as in standard scheduling,

---

[1] $\tilde{O}$ notation neglects the logarithm term in the bound.

address various machine models, different processing formats (preemptive (Chen et al., 1994) or non-preemptive (Epstein & Favrholdt, 2005)) and various objective functions (such as makespan (Graham, 1966), sum of completion times (Fiat & Woeginger, 1999), sum of flow time (Leonardi & Raz, 1997; Leonardi, 2003)). However, in previous online scheduling settings, tasks usually can wait in a line and need not to be processed immediately, and those preempted tasks also have the chance to be processed later. This is not satisfied with the streaming task setting where each task must be decided whether to process immediately, and we fill this blank in this work.

The concept of immediate decision-making has been explored in numerous online resource allocation studies, which are often motivated by real-world applications such as cloud computing platforms and E-commerce. In most of these existing works, the focus is on scenarios involving non-preemptable jobs (Gallego et al., 2015; Dickerson et al., 2019; Asadpour et al., 2020; Ekbatani et al., 2025). In such settings, once a job is assigned to a machine, it remains there until completion and cannot be preempted. The analysis in these studies differs significantly from our work. In a preemptable job scenario, decisions need to be made continuously, and new arriving jobs must be compared with currently running jobs at every time step. This makes it more challenging compared to the non-preemptable job scenarios in the existing literature.

Only a few works study online scheduling with immediate decisions and preemptable jobs, but their settings and objectives are different from ours. Canetti & Irani (1995) study the one-machine scheduling problem with a random policy. Lucier et al. (2013); Aminian et al. (2023) study the deadline setting where each job must finish before a certain time. And preempted jobs can be re-assigned before the deadline. By contrast, we study a more general multi-machine online scheduling problem, and the preempted jobs are rejected.

**Online scheduling with uncertain rewards.** There is a series of works using bandits to handle the problem of uncertain rewards in online scheduling. In Gao et al. (2021), the authors utilize a UCB variant to address the exploration-exploitation dilemma in online task scheduling. In Cayci et al. (2019), online bandit feedback is leveraged to schedule tasks in a renewal system. Xu et al. (2024) incorporates optimistic estimation for reward-to-cost ratio to learn the optimal scheduling with bandit feedback. Zhou (2024) and Wang et al. (2024) study the computational resource efficient learning where the learner allocates resources to tasks in order to complete the as many tasks as possible. We note that those works do not consider the setting of immediate decision, which is a key property in streaming data processing scenario.

## 3 SETTING

**Online Scheduling with Immediate Decision.** This paper considers an online scheduling problem consisting of $M$ machines and $K$ tasks. Each task $k$ has arriving time $b_k$, processing time $\ell_k$ (bounded with upper bound $L_{\max}$), and reward $r_k$.

There are totally $T$ rounds. At each time $t$, if a task $k$ processed by the machine satisfies $b_k + \ell_k = t$, it is completed and leaves the machine, and the learner gets the reward $r_k$. Denote $f_k = \mathbb{1}\{k \text{ is completed}\}$ as the indicator that $k$ is completed. Then a set of tasks $\mathcal{L}_t$ with $\mathcal{L}_t = \{k \mid b_k = t\}$ comes, the learner observes the set $\mathcal{L}_t$ and $\ell_k$ for each $k \in \mathcal{L}_t$. Then the learner can take one of the following actions: (a) let the empty machine (if exists) process a task $k \in \mathcal{L}_t$; (b) preempt the machine and let it process a task $k \in \mathcal{L}_t$. The preempted task $k'$ is discarded, $f_{k'} = 0$; (c) skip the task $k \in \mathcal{L}_t$, $f_k = 0$. This immediate decision setting models the real-world examples in streaming tasks applications like IoT data processing, cloud computing system and financial data processing. The objective is to maximize the total rewards of completed tasks, i.e., $\max \sum_{k=1}^{K} f_k r_k$.

Note that it is impossible to derive an optimal algorithm that achieves the maximum total rewards since the learner is unaware of the future task knowledge. Thus it is natural to analyze the competitive ratio of the proposed online algorithm ALG (Pruhs et al., 2004). Denote $\mathcal{I}$ as the set of all possible task instances. Let $R_{\text{ALG}}(I)$ denote the total reward of the completed tasks produced by the online algorithm ALG for an input instance $I \in \mathcal{I}$, and $R_{\text{OPT}}(I)$ denote the total reward of the optimal offline schedule for the same instance $I$, i.e., the optimal solution given the prior knowledge of all tasks' information. The competitive ratio $CR$ is defined as $CR = \inf_{I \in \mathcal{I}} \frac{R_{\text{ALG}}(I)}{R_{\text{OPT}}(I)}$.

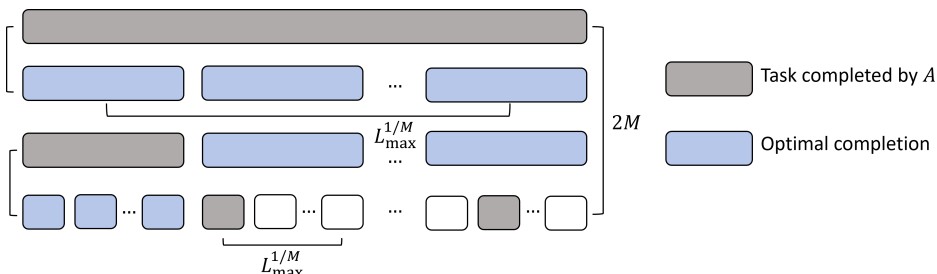

Figure 1: Illustration of worst-case instance design. There are $2M$ tasks at each time, and each two tasks form a competitive pair for $M$ machines. Suppose any algorithm $A$ completes tasks colored gray, we can always design the stream of coming tasks to construct the optimal completion colored blue with $\tilde{O}(1/(ML_{\max}^{1/M}))$ ratio.

**Stochastic Reward Model.** In most real-world online scheduling cases, the reward of a task is usually unknown and stochastic (Gao et al., 2021; Xu et al., 2024). This paper considers the stochastic reward model where there are $N$ types of tasks, and the reward of each task with type $n \in [N]$ follows a fixed and unknown 1-subgaussian distribution $\mathcal{D}_n$ with mean $\mu_n$. When task $k$ arrives, the learner can observe the task's type $n(k) \in [N]$ and its length $\ell_k$, but without observing the reward $r_k$. When the machine has completed task $k$, it receives and observes $r_k$ sampled from the fixed distribution $\mathcal{D}_{n(k)}$ with mean $\mu_{n(k)}$. Section 5 provides a thorough analysis for this uncertain reward setting.

## 4 ONLINE SCHEDULING WITH KNOWN REWARDS

In this section, we provide analysis for the setting of $M$ machines and known rewards. Section 4.1 provides worst-case competitive ratio of order $\tilde{O}(1/(ML_{\max}^{1/M}))$. Then Section 4.2 proposes a scheduling algorithm achieves $\tilde{\Omega}(1/(ML_{\max}^{1/M}))$ competitive ratio.

### 4.1 WORST-CASE COMPETITIVE RATIO ANALYSIS

In this subsection, we derive a worst-case competitive ratio of order $\tilde{O}(1/(ML_{\max}^{1/M}))$ for $M$ machines scheduling problem, shown in the following theorem.

**Theorem 4.1.** *There exists an instance containing a set of tasks such that no deterministic algorithm can achieve the competitive ratio greater than $2\log(L_{\max})/(ML_{\max}^{1/M})$.*

The intuitive idea to derive such a worst-case competitive ratio bound is that we construct competitive task pairs for each machine to select at each time $t$. Each competitive pair comprises a high-reward long task accompanied by multiple low-reward short tasks whose aggregate reward exceeds that of the long task. This construction ensures that regardless of the machine's selection, we can always generate an alternative task selection with higher total rewards.

In the multi-machine setting (with $M$ machines), the construction of competitive tasks becomes more challenging because we must prevent different machines from selecting the same competitive pair—a situation that would invalidate the competitive analysis. To overcome this, we design a competitive task set comprising $2M$ tasks at each time step, with a total of $M+1$ distinct progressing lengths ranging from 1 to $L_{\max}$. Once $M$ tasks are processed by the $M$ machines, the remaining $M$ unselected tasks form competitive pairs with them. Figure 1 provides the illustration of our instance design. A key innovation lies in the fact that our instance design guarantees the $M$ chosen tasks and the $M$ unselected tasks can be paired one-to-one, with each pair consisting of two different lengths' tasks. This length difference ensures the success of the optimal competitive analysis. The ratio between successive task lengths is carefully set as $1/L_{\max}^{1/M}$, which leads to the desired competitive ratio bound. The complete proof is deferred in Appendix A.

## 4.2 MAXIMUM REMAINING DENSITY FIRST ALGORITHM

In this subsection, we provide the algorithm for the scheduling problem with $M$ machines, attaining a competitive ratio at least $\tilde{\Omega}(1/(ML_{\max}^{1/M}))$ for any instance.

Intuitively, the algorithm assigns machines with different responsibilities. Specifically, the $i$-th machine ($1 \leq i \leq M$) only processes the task with length between $L_{\max}^{\frac{i-1}{M}}$ and $L_{\max}^{\frac{i}{M}}$. This design guarantees the length ratio between tasks for each machine is upper bounded by $L_{\max}^{\frac{1}{M}}$.

---

**Algorithm 1** Maximum Remaining Density First (for $i$-th machine)

**Input:** $L_{\max}, M$.
1: Initialize: $\mathcal{M} = \emptyset$;
2: **for** $t = 1, 2, \cdots$ **do**
3:     **if** $\mathcal{M} \neq \emptyset$ and $b_k + \ell_k = t, k \in \mathcal{M}$ **then**
4:         Receive reward $r_k$;
5:         $\mathcal{M} = \emptyset$;
6:     **end if**
7:     Receive set $\mathcal{L}_t = \{k \mid b_k = t, L_{\max}^{\frac{i-1}{M}} < \ell_k \leq L_{\max}^{\frac{i}{M}}\}$, observe $b_k, \ell_k, r_k$ for each $k \in \mathcal{L}_t$;
8:     **if** $\mathcal{M} \neq \emptyset$ and $k \in \mathcal{M}$ satisfies $\ell_k - (t - b_k) > L_{\max}^{\frac{i-1}{M}}$ **then**
9:         Construct a pseudo task $k'$ with $b_{k'} = t, \ell_{k'} = \ell_k - (t - b_k), r_{k'} = r_k$;
10:        $\mathcal{L}_t = \mathcal{L}_t \cup \{k'\}$;
11:     **end if**
12:     Find task $k \in \arg\max_{k' \in \mathcal{L}_t} \frac{r_{k'}}{\ell_{k'}}$;
13:     **if** $\mathcal{M} \neq \emptyset$ and $k' \in \mathcal{M}$ satisfies $\ell_{k'} - (t - b_{k'}) \leq L_{\max}^{\frac{i-1}{M}}$ **then**
14:         **if** $r_k > L_{\max}^{\frac{1}{M}} r_{k'}$ **then**
15:             $k = k'$
16:         **end if**
17:     **end if**
18:     $\mathcal{M} = \{k\}$;
19: **end for**

---

The algorithm design for the $i$-th machine plays the role of selecting the maximum remaining density (Line 7 - 12). This ensures the reward of the missed task will not exceed $L_{\max}^{\frac{1}{M}}$ times the processing task since the length ratio is bounded.

For the processed task with remaining length less than $L_{\max}^{\frac{i-1}{M}}$, we add "protect scheme", which only switches to a new task when that task has more than $L_{\max}^{\frac{1}{M}}$ times the reward (Line 13 - 17). This design guarantees that the algorithm would not suffer a reward loss over $1/L_{\max}^{\frac{1}{M}}$, which is a key part in the competitive ratio analysis.

**Theorem 4.2.** *For any instance $\mathcal{I}$, the max remaining density first algorithm with $M$ machines can achieve the competitive ratio greater than $1/(3ML_{\max}^{\frac{1}{M}})$.*

Proof sketch: The analysis separates the tasks processed by each machine $i \in [M]$. For every completed task $k$ with $L_{\max}^{\frac{i-1}{M}} < \ell_k \leq L_{\max}^{\frac{i}{M}}$, we compare it with the nearest optimal completed task in the $i$-th machine. Specifically, those tasks are divided into three sets by their completion times (before $k$ arrives, before $k$ completes, and after $k$ completes). Every optimal task set guarantees that its total rewards do not exceed $L_{\max}^{1/M} r_k$. Since there are at most $M$ optimal tasks at the same time, the corresponding ratio is bounded by $1/(3ML_{\max}^{\frac{1}{M}})$. The complete proof is deferred to Appendix B.

*Remark* 4.3. Note that constructing a pseudo task $k'$ for the processing task $k \in \mathcal{M}$ (Line 9 - 10) is a crucial step in algorithm design and is key to obtaining $\Omega(1/(ML_{\max}^{1/M}))$ competitive ratio. This design ensures the maximum remaining density will increase with time until the machine completes the task. Even though the machine might switch to other tasks, its selecting threshold $\max \frac{r}{\ell}$ will keep increasing, which is crucial to upper bound the summation reward of the optimal schedule by bounding the maximum density at each time.

# 5 ONLINE SCHEDULING WITH RANDOM AND UNKNOWN REWARDS

---

**Algorithm 2** Scheduling Upper Confidence Bound (S-UCB)

---

1: Initialize: $\text{UCB}_n = \infty, \mathcal{M} = \emptyset, T_n = 0$;
2: **for** $t = 1, 2, \cdots$ **do**
3:     **if** $\mathcal{M} \neq \emptyset$ and $b_k + \ell_k = t, k \in \mathcal{M}$ **then**
4:         Receive reward $r_k$, update $\text{UCB}_{n(k)}$, and $T_{n(k)}$;
5:         $\mathcal{M} = \emptyset$;
6:     **end if**
7:     Receive coming tasks set $\mathcal{L}_t = \{k \mid b_k = t, L_{\max}^{\frac{i-1}{M}} < \ell_k \leq L_{\max}^{\frac{i}{M}}\}$, observe $b_k, \ell_k, n(k)$ for each $k \in \mathcal{L}_t$;
8:     **if** $\mathcal{M} \neq \emptyset$ and $k \in \mathcal{M}$ satisfies $\ell_k - (t - b_k) > L_{\max}^{\frac{i-1}{M}}$ **then**
9:         For $k \in \mathcal{M}$, construct pseudo task $k'$ with $b_{k'} = t, \ell_{k'} = \ell_k - (t - b_k), n(k') = n(k)$;
10:       $\mathcal{L}_t = \mathcal{L}_t \cup \{k'\}$;
11:     **end if**
12:     Find task $k \in \arg\max_{k' \in \mathcal{L}_t} \frac{\text{UCB}_{n(k')}}{\ell_{k'}}$;
13:     **if** $\mathcal{M} \neq \emptyset$ and $k' \in \mathcal{M}$ satisfies $\ell_{k'} - (t - b_{k'}) \leq L_{\max}^{\frac{i-1}{M}}$ **then**
14:         **if** $\text{UCB}_{n(k)} > L_{\max}^{\frac{1}{M}}\text{UCB}_{n(k')}$ **then**
15:            $k = k'$
16:         **end if**
17:     **end if**
18:     $\mathcal{M} = \{k\}$;
19: **end for**

---

In this section, we provide the Scheduling Upper Confidence Bound (S-UCB) algorithm for online scheduling with an unknown and random rewards model introduced in Section 3.

To measure the performance of an online algorithm ALG with total rewards $R_{\text{ALG}}(T)$, it is natural to compare it with the total rewards obtained by running the optimal scheduling strategy. However, since the optimal strategy is impossible to reach (shown in Theorem 4.1) and the current best competitive ratio we can get is $1/(3ML_{\max}^{1/M})$, obtained by Algorithm 1 (shown in Theorem 4.2). We define the corresponding $1/(3ML_{\max}^{1/M})$-approximate regret, which is defined as the expectation of difference between cumulative rewards the algorithm obtains and $1/(3ML_{\max}^{1/M})$ times the optimal cumulative rewards $R_{\text{OPT}}(T)$. Formally,

$$\text{Reg}(T) = \mathbb{E}\left[\frac{1}{3ML_{\max}^{1/M}}R_{\text{OPT}}(T) - R_{\text{ALG}}(T)\right],$$

where the expectation is taken over the randomness of rewards.

The key idea in the S-UCB algorithm is that the algorithm utilizes the upper confidence bound (UCB) index for each type of task to balance exploration and exploitation. At each time, the algorithm performs the MRDF algorithm (Algorithm 1) while replacing the reward $r_k$ of each task $k$ by its UCB index $\text{UCB}_{n(k)}$, where $n(k)$ is the type of task $k$. The UCB index $\text{UCB}_n$ of each type $n \in [N]$ is computed as $\text{UCB}_n = \hat{\mu}_n + \sqrt{\frac{6\log T}{T_n}}$, where $\hat{\mu}_n$ is the empirical reward mean of task $n$ and $T_n$ is the number of completions for tasks with type $n$.

Intuitively, the UCB index of a type $n$ consists of the exploitation term $\hat{\mu}_n$ and the exploration term $\sqrt{\frac{6\log T}{T_n}}$, which is also noted as the confidence radius. This design corresponds to the fact that the reward mean $\mu_n$ of type $n$ is upper bounded by $\text{UCB}_n$ with high probability $O(1/T)$. Thus the UCB index is an appropriate statistic to balance the exploration and exploitation.

Below, we provide the regret analysis for the S-UCB algorithm.

The key technical challenge in deriving a regret upper bound beyond traditional UCB analysis lies in the preemptable property of each machine. Specifically, when machine $i \in [M]$ selects a job $k$

with the highest UCB density $\mathrm{UCB}_{n(k)}/\ell_k$ at time $t$, the reward of $k$ may not be realized since it may be preempted by another job with a higher UCB density. This makes the decision to select task $k$ at time $t$ without receiving any feedback, leading to the failure to update $\mathrm{UCB}_{n(k)}$.

To solve this issue, we combine the standard UCB analysis with the monotone property of the MRDF algorithm. Specifically, we analyze the regret of each completed task $k$. The main challenge is to upper bound the regret of those preempted jobs before $k$. Since MRDF guarantees the increase of density with time, its UCB index is also increasing with time in the S-UCB algorithm. Thus the final completed task $k$ has the highest UCB density among all previous preempted tasks. Additionally, the completion of task $k$ provides the scheduler the realization of reward $r_k$, which is used to update $\mathrm{UCB}_{n(k)}$ and serves as the key to upper bound the exploration times.

**Theorem 5.1.** *When the scheduler follows the S-UCB algorithm (Algorithm 2) with $N$ task types, the learner can obtain the $\frac{1}{3ML_{\max}^{1/M}}$-approximate regret bounded by*

$$\mathrm{Reg}(T) \leq O\left(\frac{N^2 L_{\max} \log T}{\Delta}\right),$$

*where $\Delta = \min_{n,n',\ell,\ell',\frac{\mu_n}{\ell} - \frac{\mu_{n'}}{\ell'} > 0} \frac{\mu_n}{\ell} - \frac{\mu_{n'}}{\ell'}$ is the minimum density gap of two tasks.*

*Proof.* We first introduce some notations. Denote $\mathcal{K}_{\mathrm{ALG}}^i$ as the set of completed tasks for machine $i$ by running the S-UCB. And $\mathcal{K}_{\mathrm{OPT}}^i$ is the set of optimal completed tasks with length $(L_{\max}^{\frac{i-1}{M}}, L_{\max}^{\frac{i}{M}}]$. For each $k \in \mathcal{K}_{\mathrm{ALG}}^i$, we denote $\mathcal{N}_k = \{k' \in \mathcal{K}_{\mathrm{OPT}}^i : b_{k-1} + \ell_{k-1} \leq b_{k'} < b_k + \ell_k\}$ as the set of optimal tasks that is around $k$, where $k-1$ is the nearest algorithm completed task before $k$. Denote $\mathcal{G}_k := \forall k' \in \mathcal{N}_k, \mu_{n(k)}/\ell_k \geq \mu_{n(k')}/\ell_{k'}$ as the event that completed task $k$ is truly the task with the highest reward density among all optimal tasks nearby. The regret can be rewritten as

$$\mathrm{Reg}(T) = \mathbb{E}\left[\frac{1}{3ML_{\max}^{1/M}} \sum_{i=1}^{M} \sum_{k \in \mathcal{K}_{\mathrm{ALG}}^i} \left(\sum_{k' \in \mathcal{N}_k} r_{k'} - r_k\right)\right]$$

$$= \mathbb{E}\left[\frac{1}{3ML_{\max}^{1/M}} \sum_{i=1}^{M} \sum_{k \in \mathcal{K}_{\mathrm{ALG}}^i} \mathbb{1}\{\mathcal{G}_k\} \left(\sum_{k' \in \mathcal{N}_k} r_{k'} - r_k\right)\right] + \mathbb{E}\left[\frac{1}{3ML_{\max}^{1/M}} \sum_{i=1}^{M} \sum_{k \in \mathcal{K}_{\mathrm{ALG}}^i} \mathbb{1}\{\neg\mathcal{G}_k\} \left(\sum_{k' \in \mathcal{N}_k} r_{k'} - r_k\right)\right],$$

where $\mathcal{G}_k$ decomposes the regret into two terms: the first is the regret caused by the exploitation (accurate estimation of UCB index), and the second term is the regret caused by exploration (necessary selection to reduce sub-optimal UCB density).

To bound the regret caused by the exploitation, we first state that when $\exists k, \forall k' \neq k, \frac{\mathrm{UCB}_{n(k)}}{\ell_k} > \frac{\mathrm{UCB}_{n(k')}}{\ell_{k'}}$ implies $\frac{\mu_{n(k)}}{\ell_k} > \frac{\mu_{n(k')}}{\ell_{k'}}$, the S-UCB algorithm proceeds the same as the MRDF with known mean reward $\mu_n$. This is because the S-UCB makes the exact same decision as MRDF at each time $t$. Then under event $\mathcal{G}_k$ for completed task $k$, S-UCB shares the same competitive ratio property as MRDF since $k$ is actually the task with the highest density. This indicates that the algorithm achieves the competitive ratio larger than $1/(3ML_{\max}^{1/M})$ when UCB index is well estimated, i.e., $\mathcal{G}_k$ holds, and the regret is 0.

**Lemma 5.2.** *The regret caused by exploiting the UCB index in S-UCB algorithm is bounded by*

$$\mathbb{E}\left[\frac{1}{3ML_{\max}^{1/M}} \sum_{i=1}^{M} \sum_{k \in \mathcal{K}_{\mathrm{ALG}}^i} \mathbb{1}\{\mathcal{G}_k\} \left(\sum_{k' \in \mathcal{N}_k} r_{k'} - r_k\right)\right] = 0.$$

Then we turn to bound the second term, which is the regret caused by exploring sub-optimal UCB density. This term is upper bounded by regret of selecting sub-optimal UCB density times the number of explorations for each sub-optimal selection. The following lemma upper bounds the exploration regret for each task. It should be noted that this lemma is technically non-trivial since it integrates the monotone increasing density property of the MRDF algorithm into the analysis of UCB estimation. This makes the success of deriving an optimal gap-dependent regret upper bound $O(\log T/\Delta)$.

**Lemma 5.3.** *The regret caused by exploring the UCB index in S-UCB algorithm is bounded by*

$$\mathbb{E}\left[\frac{1}{3ML_{\max}^{1/M}}\sum_{i=1}^{M}\sum_{k\in\mathcal{K}_{ALG}^{i}}\mathbb{1}\{\neg\mathcal{G}_k\}\,(\sum_{k'\in\mathcal{N}_k}r_{k'}-r_k)\right]\leq\sum_{n\in[N]}O\left(\frac{NL_{\max}\log T}{\Delta_n}\right)\leq O\left(\frac{N^2L_{\max}\log T}{\Delta}\right),$$

*where $\Delta_n=\min_{n'\neq n,\ell,\ell',\frac{\mu_{n'}}{\ell}-\frac{\mu_n}{\ell'}>0}\frac{\mu_{n'}}{\ell}-\frac{\mu_n}{\ell'}$ is the minimum density gap among task $n$.*

Then summing over these two terms, Theorem 5.1 is derived. $\qquad\square$

The following remark discusses the regret optimality of our algorithm.

*Remark* 5.4. Note that an $O(N^2L_{\max}\log T/\Delta)$ regret upper bound is optimal with respect to term $T$ in the bandit literature (Lattimore & Szepesvári, 2020). To distinguish two densities of tasks, it needs $\Omega(\log T/\Delta^2)$ times of exploration, which is unavoidable in regret analysis. Furthermore, each time selecting sub-optimal tasks incurs at least $L_{\max}\Delta$ reward gap, which is also necessary since one can construct a task instance where all tasks have length $L_{\max}$. Thus, by reducing the problem to multiple machines with equal task lengths, we can obtain an $\Omega(\frac{NL_{\max}\log T}{\Delta})$ regret lower bound, indicating our algorithm is optimal with $T,\Delta,L_{\max}$. It remains open to design an algorithm achieving better dependence on parameter $N$ and exploring possible dependence on $M$. We leave it as an interesting future work.

In addition, we also provide the competitive ratio result for the S-UCB algorithm.

**Theorem 5.5.** *Given instance I, following the S-UCB algorithm (Algorithm 2), the scheduler can achieve the expected competitive ratio $\mathbb{E}\left[CR(I)\right]=\mathbb{E}\left[R_{OPT}(I)/R_{ALG}(I)\right]$ with*

$$\mathbb{E}\left[CR(I)\right]\geq\frac{1}{3ML_{\max}^{1/M}+24N^2L_{\max}\log T/(\mathbb{E}\left[R_{ALG}(I)\right]\Delta)},$$

*where $R_{ALG}(I)$ is the total completion rewards obtained by S-UCB, and expectation is taken over the randomness of reward distribution.*

Moreover, if there are new tasks continuously coming in every round $t$, the competitive ratio has the bound of $1/\left(3ML_{\max}^{1/M}+N^2L_{\max}\log T/(T\Delta^2)\right)$. This theorem states that the expected competitive ratio will asymptotically converge to $1/(3ML_{\max}^{1/M})$ when $T$ increases, indicating that S-UCB will perform the same as MRDF when UCB index is well estimated.

## 6 EXPERIMENTS

In this section, we conduct experiments to show the efficiency of our proposed MRDF algorithm (Algorithm 1) and S-UCB algorithm (Algorithm 2).

**Setting.** We take $T=500,000$ and $M=5$. $N$ is set as 10 and the reward of each type $n$ follows a Gaussian distribution $\mathcal{N}(\mu_n,\sigma^2)$ with $\sigma=1$, where $\mu_n$ is uniformly selected from $[0,1]$. $L_{\max}$ is set as 30 and each task's length is sampled uniformly from $[1,L_{\max}]$. Each experiment is repeated 50 times. All plots are averaged over 50 trials with confidence intervals of 95%.

**Baselines.** We first compare the MRDF algorithm (Algorithm 1) to other baseline algorithms in the setting of known reward. Specifically, we consider the following representative baselines to justify MRDF's performance:

- Maximum reward (MR), which always processes the task with the highest reward.
- Shortest remaining processing time (SRPT), which always processes the task with the minimum remaining processing time.
- Optimal scheduling strategy (OPT), which needs prior knowledge of reward and processing time information for all tasks before the game. The optimal strategy can be computed by dynamic programming.

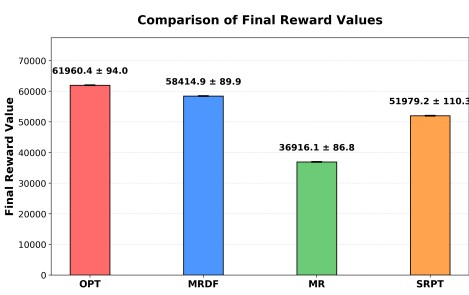

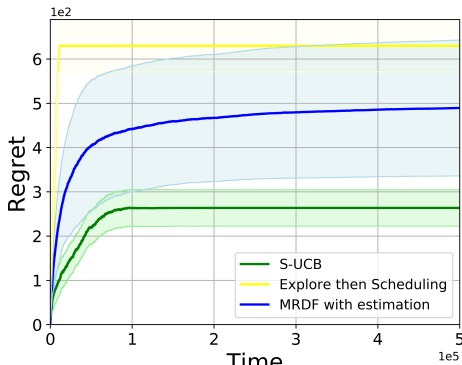

Figure 2: Cumulative rewards for the optimal scheduling strategy, MRDF algorithm, MR algorithm, and SRPT algorithm under the setting of known reward.

Figure 3: Regret for S-UCB algorithm (Algorithm 2) and baselines under the setting of random and unknown reward.

In the setting of known reward, the scheduler can observe the sampled reward when the task comes. Figure 2 shows the comparisons of cumulative rewards for MRDF algorithm with other baseline algorithms. It can be seen that our MRDF algorithm shows the comparable performance with the optimal strategy when the processing time is uniformly distributed, and it beats other baseline algorithms. This result shows the effectiveness of our proposed MRDF algorithm. Note that when the processing time is uniformly distributed, which usually happens in real-world application, the worst case occur with low probability. Thus the MRDF algorithm can achieve the competitive ratio larger than $1/(3ML_{\max}^{1/M})$.

Then we test the efficiency and convergence of our S-UCB algorithm (Algorithm 2) under the setting of uncertain rewards. We compare it with two baseline algorithms: Explore then Scheduling and MRDF with estimation. Explore then Scheduling algorithm first explores all types uniformly until it is confident about the estimation, and then it uses this estimation to run the MRDF algorithm (algorithm's detail is provided in Appendix E, and we also provide the regret analysis for this algorithm, which can only obtain the $O(\log T/\Delta^2)$ regret). MRDF with estimation algorithm always executes MRDF with the current reward estimations instead of the UCB index. We plot the regret curve of these algorithms in Figure 3. It can be seen that the regret curve of S-UCB is sub-linear with time horizon, and S-UCB also has lower regret and better stability than other two baselines. This verifies the result in Theorem 5.1 and shows the convergence and effectiveness of our algorithm.

## 7 CONCLUSION

Our study on online scheduling with immediate decisions and uncertain rewards, motivated by streaming task applications, has yielded significant results. By initially focusing on the case of known rewards, we are able to establish the worst-case competitive ratio of $\tilde{O}(1/(ML_{\max}^{1/M}))$, which provides a clear benchmark for understanding the performance limits of the scheduling problem under such conditions. The proposed near-optimal scheduling MRDF (Algorithm 1) for this scenario achieves the competitive ratio of $\Omega(1/(ML_{\max}^{1/M}))$. When it comes to the more challenging situation of random and unknown rewards, our proposed S-UCB (Algorithm 2) algorithm attains an $O(\log T/\Delta)$ regret, showcasing its effectiveness in handling the inherent uncertainty. Additionally, we also provide the competitive ratio analysis for the S-UCB algorithm. Extensive experiments compared with baselines also show the effectiveness of our proposed algorithms.

**Future Work**   In this work, we study the competitive ratio for immediate decision setting under deterministic policy. An interesting future direction is to extend the analysis to random policy, which attains target objective in expectation or with high probability. Another future direction is to improve the existing regret bound for stochastic reward setting and extend the analysis to other online scheduling with uncertain rewards problems.

**Ethics statement**    This paper presents work whose goal is to advance the field of Online Learning. There are many potential societal consequences of our work, none which we feel must be specifically highlighted here.

**Reproducibility statement**    The detail and description of the experiments are provided in experiments section. We also provide the complete code to reproduce.

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

## A   PROOF OF THEOREM 4.1

The task set is constructed as follows:

For the task 0: $b_0 = 1, \ell_0 = L_{\max}, r_0 = L_{\max}$.

For the task 1: $b_1 = 1, \ell_1 = 1, r_1 = 1$.

For the task $i$ with $2 \leq i \leq L_{\max}$: $b_i = i, \ell_i = 1, r_i = \frac{2 \log L_{\max}}{L_{\max}} \sum_{j=1}^{i-1} r_j$.

Then consider any deterministic scheduling algorithm $\mathcal{A}$. If the algorithm $\mathcal{A}$ selects 1 at time 1, consider another task instance which has only tasks $0, 1$, the ratio it gets is $1/L_{\max} < 2 \log(L_{\max})/L_{\max}$.

If $\mathcal{A}$ selects 0 at first but switches to $i$ with $2 \leq i \leq L_{\max}$, we consider the task instance which only has $0, \cdots, i$, and the rewards $\mathcal{A}$ loses is at least $\sum_{j=1}^{i-1} r_j$, and the corresponding competitive ratio $\mathcal{A}$ gets is at most $r_i / \sum_{j=1}^{i-1} r_j = 2 \log(L_{\max})/L_{\max}$.

If $\mathcal{A}$ selects 0 and completes it, the maximum total reward is by selecting tasks $1, \cdots, L_{\max}$. Since $r_i = \frac{2 \log(L_{\max})}{L_{\max}} \sum_{j=1}^{i-1} r_j$, we have that

$$\sum_{j=1}^{i} r_j = \left(1 + \frac{2 \log L_{\max}}{L_{\max}}\right) \sum_{j=1}^{i-1} r_j = \left(1 + \frac{2 \log L_{\max}}{L_{\max}}\right)^{i-1}.$$

The competitive ratio $\mathcal{A}$ achieves is

$$\frac{r_0}{\sum_{i=1}^{L_{\max}} r_i} = \frac{L_{\max}}{(1 + \frac{2 \log L_{\max}}{L_{\max}})^{L_{\max}-1}} < \frac{2 \log(L_{\max})}{L_{\max}},$$

the inequality holds when $L_{\max} \geq 6$.

All possible behaviors of the algorithm $\mathcal{A}$ have been analyzed, and no one can achieve the ratio larger than $\frac{2 \log L_{\max}}{L_{\max}}$. Thus any deterministic can not achieve the competitive ratio larger than $\frac{2 \log(L_{\max})}{L_{\max}}$.

*Proof.* The instance $I$ is designed as follows:

- For the task with length $L_{\max}$, we design a task with $b_{M,1} = 1, \ell_{M,1} = L_{\max}, r_{M,1} = L_{\max}$.

- For the task with length $L_{\max}^{\frac{M-1}{M}}$, we design $2L_{\max}^{\frac{1}{M}}$ tasks, indexed by $(M-1, i, j)$, $i \in [L_{\max}^{\frac{1}{M}}], j \in \{0,1\}$. For $i = 1$: $b_{M-1,1,0} = b_{M-1,1,1} = 1$, $\ell_{M-1,1,0} = \ell_{M-1,1,1} = L_{\max}^{\frac{M-1}{M}}$, $r_{M-1,1,0} = r_{M-1,1,1} = L_{\max}^{\frac{M-1}{M}}$. For $i > 1$: $b_{M-1,i,0} = b_{M-1,i,1} = (i-1)L_{\max}^{\frac{M-1}{M}} + 1$, $\ell_{M-1,i,0} = \ell_{M-1,i,1} = L_{\max}^{\frac{M-1}{M}}$, $r_{M-1,i,0} = r_{M-1,i,1} = \frac{2\log L_{\max}^{1/M}}{L_{\max}^{1/M}} \sum_{i'=1}^{i-1} r_{M-1,i',0}$.

- For the task with length $L_{\max}^{\frac{m}{M}}$, $m \geq 1$, we design $2L_{\max}^{\frac{M-m}{M}}$ tasks, indexed by $(m, i, j)$, $i \in [L_{\max}^{\frac{M-m}{M}}], j \in \{0,1\}$. For $i = kL_{\max}^{\frac{1}{M}} + 1$, $0 \leq k < M - m$, $b_{m,i,0} = b_{m,i,1} = (i-1)L_{\max}^{\frac{m}{M}} + 1$, $\ell_{m,i,0} = \ell_{m,i,1} = L_{\max}^{\frac{m}{M}}$, $r_{m,i,0} = r_{m,i,1} = r_{m+1,k,0}/L_{\max}^{\frac{1}{M}}$. For $i = kL_{\max}^{\frac{1}{M}} + n$, $1 < n \leq L_{\max}^{\frac{1}{M}}$. $b_{m,i,0} = b_{m,i,1} = (i-1)L_{\max}^{\frac{m}{M}} + 1$, $\ell_{m,i,0} = \ell_{m,i,1} = L_{\max}^{\frac{m}{M}}$, $r_{m,i,0} = r_{m,i,1} = \frac{2\log L_{\max}^{1/M}}{L_{\max}^{1/M}} \sum_{n'=1}^{n-1} r_{m,kL_{\max}^{1/M}+n',0}$.

- For the task with length $1$, we design $L_{\max}$ tasks, indexed by $(1, i)$, $i \in [L_{\max}]$. For $i = kL_{\max}^{\frac{1}{M}} + 1$, $0 \leq k < M$, $b_{1,i} = i$, $\ell_{1,i} = 1$, $r_{1,i} = r_{2,k,0}/L_{\max}^{\frac{1}{M}}$. For $i = kL_{\max}^{\frac{1}{M}} + n$, $1 < n \leq L_{\max}^{\frac{1}{M}}$. $b_{1,i} = i$, $\ell_{1,i} = 1$, $r_{1,i} = \frac{2\log L_{\max}^{1/M}}{L_{\max}^{1/M}} \sum_{n'=1}^{n-1} r_{1,kL_{\max}^{1/M}+n'}$.

Consider any deterministic scheduling algorithm $\mathcal{A}$. At time $t = 1$ algorithm $\mathcal{A}$ selects $M$ tasks among these $2M$ tasks. Sort and index the selected tasks by increasing order of length. Similarly, those unselected tasks are also sorted by the increasing order. And thus the selected task $i_k, k \in [M]$ competes with the unselected task $i'_k$. By the instance design it holds that $\ell_{i_k} \neq \ell_{i'_k}, \forall k \in [M]$. If $\ell_{i_k} < \ell_{i'_k}$, we construct another instance $\mathcal{I}'$ such that task $i_k$ does not appear until task $\ell_{i'_k}$ ends. The competitive ratio of it is larger than $1/L_{\max}^{1/M}$. If $\ell_{i_k} > \ell_{i'_k}$, the instance $\mathcal{I}'$ receives task $\ell_{i'_k}$ as $\mathcal{I}$ does.

If $\mathcal{A}$ switches its selected tasks at some time $t$. It must switch task $i$ with length $\ell_i$ to task $i'$ with shorter length $\ell'_i$. Since the algorithm does not select tasks with length $\ell_{i'}$ before $t$, the ratio it loses for task $i'$ is

$$\frac{r_{i'}}{\sum_{j=1}^{i'-1} r_j} = \frac{2\log L_{\max}}{ML_{\max}^{1/M}}.$$

After that, the algorithm resorts the selected tasks and unselected tasks, and then pairs the compete tasks to construct the coming tasks for another instance $\mathcal{I}'$.

If $\mathcal{A}$ completes a task $i_k$ and the competitive task $i'_k$ has shorter length. The loses of rewards is the summation of those shorter tasks, which induces the competitive ratio no larger than

$$\frac{r_{i,k}}{\sum_{i'=1}^{L_{\max}^{1/M}} r_{i'}} \leq \frac{2\log L_{\max}^{1/M}}{L_{\max}^{1/M}} = \frac{2\log L_{\max}}{ML_{\max}^{1/M}}.$$

Thus we have enumerated all possible actions of the algorithm $\mathcal{A}$, and it can not achieve the competitive ratio larger than $\frac{2\log L_{\max}}{ML_{\max}^{1/M}}$.

Then we turn to a more complicated $M$-machine setting.

**Theorem A.1.** *There exists an instance containing a set of tasks such that no deterministic algorithm can achieve the competitive ratio greater than $2\log(L_{\max})/ML_{\max}^{1/M}$.*

For the simplicity suppose $L_{\max}^{1/M}$ is an integer. The instance $I$ is designed as follows:

- For the task with length $L_{\max}$, we design a task with $b_{M,1} = 1, \ell_{M,1} = L_{\max}, r_{M,1} = L_{\max}$.

- For the task with length $L_{\max}^{\frac{M-1}{M}}$, we design $2L_{\max}^{\frac{1}{M}}$ tasks, indexed by $(M-1, i, j)$, $i \in [L_{\max}^{\frac{1}{M}}], j \in \{0, 1\}$. For $i = 1$: $b_{M-1,1,0} = b_{M-1,1,1} = 1$, $\ell_{M-1,1,0} = \ell_{M-1,1,1} = L_{\max}^{\frac{M-1}{M}}$, $r_{M-1,1,0} = r_{M-1,1,1} = L_{\max}^{\frac{M-1}{M}}$. For $i > 1$: $b_{M-1,i,0} = b_{M-1,i,1} = (i-1)L_{\max}^{\frac{M-1}{M}} + 1$, $\ell_{M-1,i,0} = \ell_{M-1,i,1} = L_{\max}^{\frac{M-1}{M}}$, $r_{M-1,i,0} = r_{M-1,i,1} = \frac{2 \log L_{\max}^{1/M}}{L_{\max}^{1/M}} \sum_{i'=1}^{i-1} r_{M-1,i',0}$.

- For the task with length $L_{\max}^{\frac{m}{M}}$, $m \geq 1$, we design $2L_{\max}^{\frac{M-m}{M}}$ tasks, indexed by $(m, i, j)$, $i \in [L_{\max}^{\frac{M-m}{M}}], j \in \{0, 1\}$. For $i = kL_{\max}^{\frac{1}{M}} + 1$, $0 \leq k < M - m$, $b_{m,i,0} = b_{m,i,1} = (i-1)L_{\max}^{\frac{m}{M}} + 1$, $\ell_{m,i,0} = \ell_{m,i,1} = L_{\max}^{\frac{m}{M}}$, $r_{m,i,0} = r_{m,i,1} = r_{m+1,k,0}/L_{\max}^{\frac{1}{M}}$. For $i = kL_{\max}^{\frac{1}{M}} + n$, $1 < n \leq L_{\max}^{\frac{1}{M}}$. $b_{m,i,0} = b_{m,i,1} = (i-1)L_{\max}^{\frac{m}{M}} + 1$, $\ell_{m,i,0} = \ell_{m,i,1} = L_{\max}^{\frac{m}{M}}$, $r_{m,i,0} = r_{m,i,1} = \frac{2 \log L_{\max}^{1/M}}{L_{\max}^{1/M}} \sum_{n'=1}^{n-1} r_{m,kL_{\max}^{1/M}+n',0}$.

- For the task with length $1$, we design $L_{\max}$ tasks, indexed by $(1, i)$, $i \in [L_{\max}]$. For $i = kL_{\max}^{\frac{1}{M}} + 1$, $0 \leq k < M$, $b_{1,i} = i$, $\ell_{1,i} = 1$, $r_{1,i} = r_{2,k,0}/L_{\max}^{\frac{1}{M}}$. For $i = kL_{\max}^{\frac{1}{M}} + n$, $1 < n \leq L_{\max}^{\frac{1}{M}}$. $b_{1,i} = i$, $\ell_{1,i} = 1$, $r_{1,i} = \frac{2 \log L_{\max}^{1/M}}{L_{\max}^{1/M}} \sum_{n'=1}^{n-1} r_{1,kL_{\max}^{1/M}+n'}$.

Intuitively, by this design, we always have $2M$ tasks at every time $t \in [L_{\max}]$. Among these $2M$ tasks, every two tasks with the nearest lengths form a competitive pair. Specifically, the task with length $L_{\max}$ is challenged by tasks with length $L_{\max}^{\frac{M-1}{M}}$, and tasks with length $L_{\max}^{\frac{m}{M}}$ are challenged by tasks with length $L_{\max}^{\frac{m-1}{M}}$. Since there are $M$ machines, there are at least $M$ tasks abandoned at each time, causing the possibility to construct the instance with competitive ratio of $\log L_{\max}^{1/M}/L_{\max}^{1/M}$.

Consider any deterministic scheduling algorithm $\mathcal{A}$. At time $1$ algorithm $\mathcal{A}$ selects $M$ tasks among these $2M$ tasks. Sort and index the selected tasks by increasing order of length. Those unselected tasks are also sorted by the increasing order. And thus the selected task $i_k, k \in [M]$ competes with the unselected task $i'_k$. By the instance design we have that $\ell_{i_k} \neq \ell_{i'_k}$. If $\ell_{i_k} < \ell_{i'_k}$, we construct another instance $\mathcal{I}'$ and that task $i_k$ does not appear until task $\ell_{i'_k}$ ends. The competitive ratio of it is larger than $1/L_{\max}^{1/M}$.

If $\mathcal{A}$ switches its selected tasks at time $t$. It must switch to tasks with shorter length. It loses reward of summation of those unselected shorter task before $t$, which is $L_{\max}^{1/M}/(2 \log L_{\max}^{1/M})$ times the short task's reward, leading to a competitive ratio of $2 \log L_{\max}^{1/M}/L_{\max}^{1/M}$.

If $\mathcal{A}$ completes a task $i_k$ and the competitive task $i'_k$ has shorter length. The loses of rewards is the summation of those shorter tasks, which is at least $L_{\max}^{1/M}/(2 \log L_{\max}^{1/M})$ times $r_{i_k}$, also leading to the competitive ratio of $2 \log L_{\max}^{1/M}/L_{\max}^{1/M}$.

Thus we have enumerated all possible actions of the algorithm $\mathcal{A}$, and it can not achieve the competitive ratio larger than $2 \log L_{\max}^{1/M}/L_{\max}^{1/M} = \frac{2 \log L_{\max}}{ML_{\max}^{1/M}}$. $\qquad\square$

## B    PROOF OF THEOREM 4.2

*Proof.* We begin our analysis with one machine setting.

Denote $\mathcal{L}_{ALG}$ as the set of tasks the algorithm completes, $\mathcal{L}_{OPT}$ as the set of tasks of the optimal schedule that maximize the total reward.

For each task $k \in \mathcal{L}_{ALG}$, we analyze the optimal tasks in $\mathcal{L}_{OPT}$ related to $k$. Denote $k_{pre} \in \arg\max_{k' \in \mathcal{L}_{ALG}, b_{k'} < b_k} b_{k'}$ as the nearest completed task before $k$ by the algorithm. Then we can define the optimal task set $\mathcal{L}_{OPT,k} = \{k' \in \mathcal{L}_{OPT} : b_{k_{pre}} + \ell_{k_{pre}} \leq b_{k'} < b_k + \ell_k\}$ as the set of tasks in $\mathcal{L}_{OPT}$ such that it arrives after $k_{pre}$ is completed and before $k$ is completed. By this split it

is easy to verify that the union of $\mathcal{L}_{OPT,k}$ covers all tasks in $\mathcal{L}_{OPT}$.

$$\bigcup_{k \in \mathcal{ALG}} \mathcal{L}_{OPT,k} = \mathcal{L}_{OPT}.$$

Thus it suffices to analyze the reward relationship between $r_k$ and $\sum_{k' \in \mathcal{L}_{OPT,k}} r_{k'}$ for each task $k \in \mathcal{L}_{ALG}$.

For the set $\mathcal{L}_{OPT,k}$, we further split it into three subsets: $\mathcal{L}_{OPT,k,1} = \{k' \in \mathcal{L}_{OPT,k} : b_{k'} + \ell_{k'} \leq b_k\}$ is the set of optimal tasks completed before $k$ arrives; $\mathcal{L}_{OPT,k,2} = \{k' \in \mathcal{L}_{OPT,k} : b_{k'} \geq b_k, b_{k'} + \ell_{k'} \leq b_k + \ell_k\}$ is the set of optimal tasks arrives after $k$ arrives and completed before $k$ is completed; $\mathcal{L}_{OPT,k,3} = \{k' \in \mathcal{L}_{OPT,k} : b_{k'} + \ell_{k'} > b_k + \ell_k\}$ is the set of optimal tasks completed after $k$ is completed. Since tasks in $\mathcal{L}_{OPT}$ are disjoint, it is easy to check that

$$\mathcal{L}_{OPT,k,1} \cup \mathcal{L}_{OPT,k,2} \cup \mathcal{L}_{OPT,k,3} = \mathcal{L}_{OPT,k}.$$

We first upper bound the rewards of tasks in $\mathcal{L}_{OPT,k,1}$. Since between time $b_{k_{pre}} + \ell_{k_{pre}}$ and $b_k$ the algorithm completes no task, this indicates that the algorithm keeps processing some tasks at those times but switches to other tasks before it is completed, and in the end, the algorithm switches to task $k$ at $t = b_k$ and completes it. Denote $u_t$ as the processing task by algorithm at time $t$ and $\rho_t = r_{u_t}/(\ell_{u_t} - (t - b_{u_t}))$ as the remaining density of that processing task. By the algorithm's selecting rule, $\rho_t$ is the maximum density of all arriving tasks at time $t$. Thus the total rewards in $\mathcal{L}_{OPT,k,1}$ is maximized by

$$\sum_{k' \in \mathcal{L}_{OPT,k,1}} r_{k'} \leq \sum_{t = b_{k_{pre}} + \ell_{k_{pre}}}^{b_k - 1} \rho_t.$$

Then it suffices to upper bound $\rho_t$ at each time $t$. Since the algorithm does not complete any task at $t \in [b_{k_{pre}} + \ell_{k_{pre}}, b_k]$, the density $\rho_t$ is increasing with time $t$.

We first consider the density $\rho_t$ at $t = b_k - 1$. Recall that $u_{b_k - 1}$ is the processing task at time $t = b_k - 1$ and $\rho_t = r_{u_t}/(\ell_{u_t} - (t - b_{u_t}))$. Since the algorithm switches to $k$ at time $b_k$, it indicates that $r_k/\ell_k > r_{u_t}/(\ell_{u_t} - (t - b_{u_t}) - 1)$. Thus $\rho_t = r_{u_t}/(\ell_{u_t} - (t - b_{u_t})) < \frac{r_k}{\ell_k} \frac{\ell_{u_t} - (t - b_{u_t}) - 1}{\ell_{u_t} - (t - b_{u_t})}$, which is maximized by setting $\ell_{u_t} = L_{\max}, b_{u_t} = t$, and thus $\rho_t \leq \frac{L_{\max} - 1}{L_{\max}} \frac{r_k}{\ell_k}$ at time $t = b_k - 1$.

Then we turn to consider the density $\rho_t$ at $t = b_k - 2$. (1) If $u_{b_k - 2} = u_{b_k - 1}$, which means the algorithm does not switch at $t = b_k - 1$, and switches to task $k$ at $t = b_k$. Thus we have $\frac{r_k}{\ell_k} > \frac{r_{u_{b_k - 2}}}{\ell_{u_{b_k - 2}} - (b_k - b_{u_{b_k - 2}})}$. This is maximized by setting $b_{u_{b_k - 2}} = b_k - 2, \ell_{u_{b_k - 2}} = L_{\max}$ which implies $\rho_{b_k - 2} \leq \frac{L_{\max} - 2}{L_{\max}} \frac{r_k}{\ell_k}$. (2) If $u_{b_k - 2} \neq u_{b_k - 1}$, which means the algorithm switches to another task at $t = b_k - 1$. Then we have already know that $\rho_{b_k - 1} \leq \frac{L_{\max} - 1}{L_{\max}} \frac{r_k}{\ell_k}$, and similar analysis for $t = b_k - 2$ by setting $b_{u_{b_k - 2}} = b_k - 2, \ell_{u_{b_k - 2}} = L_{\max}$, the density is maximized by $\rho_{b_k - 2} \leq (\frac{L_{\max} - 1}{L_{\max}})^2 \frac{r_k}{\ell_k}$. Since $(\frac{L_{\max} - 1}{L_{\max}})^2 > \frac{L_{\max} - 2}{L_{\max}}$, $\rho_{b_k - 2}$ is maximized by $(\frac{L_{\max} - 1}{L_{\max}})^2 \frac{r_k}{\ell_k}$.

We prove the following bound of $\rho_t$ by induction. Suppose for any $i' \leq i$, $\rho_{b_k - i} \leq (\frac{L_{\max} - 1}{L_{\max}})^i \frac{r_k}{\ell_k}$. Then consider time $t = b_k - (i + 1)$, suppose the algorithm processes task $u_t$ at $t = b_k - (i + 1)$ and switches to another task at $t'$ with $t < t' \leq b_k$. Since the maximum density at $t'$ is $(\frac{L_{\max} - 1}{L_{\max}})^{b_k - t'} \frac{r_k}{\ell_k}$. By the selecting rule we have that $\frac{r_{u_t}}{\ell_{u_t} - (t' - b_{u_t})} < (\frac{L_{\max} - 1}{L_{\max}})^{b_k - t'} \frac{r_k}{\ell_k}$, which implies $\frac{r_{u_t}}{\ell_{u_t} - (t - b_{u_t})} <$

$\frac{\ell_{u_t} - (t' - b_{u_t})}{\ell_{u_t} - (t - b_{u_t})} \left(\frac{L_{\max} - 1}{L_{\max}}\right)^{b_k - t'} \frac{r_k}{\ell_k}$. This is maximized by setting $b_{u_t} = t, \ell_{u_t} = L_{\max}$, and we have

$$\rho_t \leq \frac{L_{\max} - (t' - t)}{L_{\max}} \left(\frac{L_{\max} - 1}{L_{\max}}\right)^{b_k - t'} \frac{r_k}{\ell_k}$$

$$= \prod_{j=1}^{t' - t} \frac{L_{\max} - j}{L_{\max} - j + 1} \left(\frac{L_{\max} - 1}{L_{\max}}\right)^{b_k - t} \frac{r_k}{\ell_k}$$

$$\leq \left(\frac{L_{\max} - 1}{L_{\max}}\right)^{t' - t} \left(\frac{L_{\max} - 1}{L_{\max}}\right)^{b_k - t'} \frac{r_k}{\ell_k}$$

$$\leq \left(\frac{L_{\max} - 1}{L_{\max}}\right)^{b_k - t} \frac{r_k}{\ell_k}$$

$$= \left(\frac{L_{\max} - 1}{L_{\max}}\right)^{i+1} \frac{r_k}{\ell_k},$$

which is maximized by switching at time $t + 1$. Thus we can conclude that for each time $t \leq b_k$, $\rho_t \leq \left(\frac{L_{\max} - 1}{L_{\max}}\right)^{b_k - t} \frac{r_k}{\ell_k}$. Thus

$$\sum_{t = b_{k_{pre}} + \ell_{k_{pre}}}^{b_k - 1} \rho_t \leq \sum_{t = b_{k_{pre}} + \ell_{k_{pre}}}^{b_k - 1} \left(\frac{L_{\max} - 1}{L_{\max}}\right)^{b_k - t} \frac{r_k}{\ell_k} \leq L_{\max} \frac{r_k}{\ell_k}.$$

This implies that

$$r_k \geq \frac{\ell_k}{L_{\max}} \sum_{t = b_{k_{pre}} + \ell_{k_{pre}}}^{b_k - 1} \rho_t \geq \frac{\ell_k}{L_{\max}} \sum_{k' \in \mathcal{L}_{OPT,k,1}} r_{k'}.$$

For optimal tasks in $\mathcal{L}_{OPT,k,2}$, the maximum total reward is obtained by setting optimal task $k'$ with $b_{k'} = t, \ell_{k'} = 1, r_{k'} = \frac{r_k}{\ell_k - (t - b_k)}$ (the maximum reward one can set when no switch happens). Then we have

$$\sum_{k' \in \mathcal{L}_{OPT,k,2}} r_{k'} \leq \sum_{t = b_k}^{b_k + \ell_k - 1} \frac{r_k}{\ell_k - (t - b_k)} \leq \ell_k r_k.$$

This indicates

$$r_k \geq \frac{1}{\ell_k} \sum_{k' \in \mathcal{L}_{OPT,k,2}} r_{k'}.$$

For optimal tasks in $\mathcal{L}_{OPT,k,3}$, we know that this set contains at most one task that is completed after $b_k + \ell_k$ (other tasks will be classified to the next task set). This optimal task $k'$ is maximized by setting $b_{k'} = b_k + \ell_k - 1, \ell_{k'} = L_{\max}, r_{k'} = L_{\max} r_k$. And thus we have

$$r_k \geq \frac{1}{L_{\max}} r_{k'}.$$

Thus together we have

$$r_k \geq \frac{\ell_k}{L_{\max}} \sum_{k' \in \mathcal{L}_{OPT,k,1}} r_{k'},$$

$$r_k \geq \frac{1}{\ell_k} \sum_{k' \in \mathcal{L}_{OPT,k,2}} r_{k'},$$

$$r_k \geq \frac{1}{L_{\max}} \sum_{k' \in \mathcal{L}_{OPT,k,3}} r_{k'},$$

which implies

$$r_k \geq \frac{1}{3L_{\max}} \sum_{k' \in \mathcal{L}_{OPT,k,1} \cup \mathcal{L}_{OPT,k,2} \cup \mathcal{L}_{OPT,k,3}} r_{k'}$$

$$= \frac{1}{3L_{\max}} \sum_{k' \in \mathcal{L}_{OPT,k}} r_{k'} .$$

Summing over all tasks $k \in \mathcal{L}_{ALG}$,

$$\sum_{k \in \mathcal{L}_{ALG}} r_k$$

$$\geq \frac{1}{3L_{\max}} \sum_{k \in \mathcal{L}_{ALG}} \sum_{k' \in \mathcal{L}_{OPT,k}} r_{k'}$$

$$= \frac{1}{3L_{\max}} \sum_{k' \in \mathcal{L}_{OPT}} r_{k'} .$$

The algorithm can achieve $1/(3L_{\max})$ competitive ratio.

Then we turn to the more complicated $M$ machines setting.

Denote $\mathcal{L}_{ALG}$ as the set of tasks the machine $i \in [M]$ completes, $\mathcal{L}_{OPT}$ as the set of tasks of the optimal schedule of tasks length between $L_{\max}^{(i-1)/M}$ and $L_{\max}^{i/M}$ that maximize the total reward.

For each task $k \in \mathcal{L}_{ALG}$, we analyze the optimal tasks in $\mathcal{L}_{OPT}$ related to $k$. Denote $k_{pre} \in \arg\max_{k' \in \mathcal{L}_{ALG}, b_{k'} < b_k} b_{k'}$ as the nearest completed task before $k$ by the algorithm. Then we can define the optimal task set $\mathcal{L}_{OPT,k} = \{k' \in \mathcal{L}_{OPT} : b_{k_{pre}} + \ell_{k_{pre}} \leq b_{k'} < b_k + \ell_k\}$ as the set of tasks in $\mathcal{L}_{OPT}$ such that it arrives after $k_{pre}$ is completed and before $k$ is completed. By this split it is easy to verify that the union of $\mathcal{L}_{OPT,k}$ covers all tasks in $\mathcal{L}_{OPT}$.

$$\bigcup_{k \in \mathcal{ALG}} \mathcal{L}_{OPT,k} = \mathcal{L}_{OPT} .$$

Thus it suffices to analyze the reward relationship between $r_k$ and $\sum_{k' \in \mathcal{L}_{OPT,k}} r_{k'}$ for each task $k \in \mathcal{L}_{ALG}$.

For the set $\mathcal{L}_{OPT,k}$, we further split it into three subsets: $\mathcal{L}_{OPT,k,1} = \{k' \in \mathcal{L}_{OPT,k} : b_{k'} + \ell_{k'} \leq b_k\}$ is the set of optimal tasks completed before $k$ arrives; $\mathcal{L}_{OPT,k,2} = \{k' \in \mathcal{L}_{OPT,k} : b_{k'} \geq b_k, b_{k'} + \ell_{k'} \leq b_k + \ell_k\}$ is the set of optimal tasks arrives after $k$ arrives and completed before $k$ is completed; $\mathcal{L}_{OPT,k,3} = \{k' \in \mathcal{L}_{OPT,k} : b_{k'} + \ell_{k'} > b_k + \ell_k\}$ is the set of optimal tasks completed after $k$ is completed. Since tasks in $\mathcal{L}_{OPT}$ are disjoint, it is easy to check that

$$\mathcal{L}_{OPT,k,1} \cup \mathcal{L}_{OPT,k,2} \cup \mathcal{L}_{OPT,k,3} = \mathcal{L}_{OPT,k} .$$

We first upper bound the rewards of tasks in $\mathcal{L}_{OPT,k,1}$. Since between time $b_{k_{pre}} + \ell_{k_{pre}}$ and $b_k$ the algorithm completes no task, this indicates that the algorithm keeps processing some tasks at those times but switches to other tasks before it is completed, and in the end, the algorithm switches to task $k$ at $t = b_k$ and completes it. Denote $u_t$ as the processing task by algorithm at time $t$ and $\rho_t = r_{u_t}/(\ell_{u_t} - (t - b_{u_t}))$ as the remaining density of that processing task. By the algorithm's selecting rule, $\rho_t$ is the maximum density of all arriving tasks at time $t$. Thus the total rewards in $\mathcal{L}_{OPT,k,1}$ is maximized by

$$\sum_{k' \in \mathcal{L}_{OPT,k,1}} r_{k'} \leq \sum_{t=b_{k_{pre}}+\ell_{k_{pre}}}^{b_k-1} \rho_t .$$

Then it suffices to upper bound $\rho_t$ at each time $t$. Since the algorithm does not complete any task at $t \in [b_{k_{pre}} + \ell_{k_{pre}}, b_k]$, the density $\rho_t$ is increasing with time $t$.

We first consider the density $\rho_t$ at $t = b_k - 1$. Recall that $u_{b_k-1}$ is the processing task at time $t = b_k - 1$ and $\rho_t = r_{u_t}/(\ell_{u_t} - (t - b_{u_t}))$. Since the algorithm switches to $k$ at time $b_k$, it indicates

that $r_k/\ell_k > r_{u_t}/(\ell_{u_t} - (t - b_{u_t}) - 1)$. Thus $\rho_t = r_{u_t}/(\ell_{u_t} - (t - b_{u_t})) < \frac{r_k}{\ell_k} \frac{\ell_{u_t} - (t - b_{u_t}) - 1}{\ell_{u_t} - (t - b_{u_t})}$, which

is maximized by setting $\ell_{u_t} = L_{\max}^{1/M}$, $b_{u_t} = t$, and thus $\rho_t \leq \frac{L_{\max}^{1/M} - 1}{L_{\max}^{1/M}} \frac{r_k}{\ell_k}$ at time $t = b_k - 1$.

Then we turn to consider the density $\rho_t$ at $t = b_k - 2$. (1) If $u_{b_k - 2} = u_{b_k - 1}$, which means the algorithm does not switch at $t = b_k - 1$, and switches to task $k$ at $t = b_k$. Thus we have $\frac{r_k}{\ell_k} > \frac{r_{u_{b_k - 2}}}{\ell_{u_{b_k - 2}} - (b_k - b_{u_{b_k - 2}})}$. This is maximized by setting $b_{u_{b_k - 2}} = b_k - 2$, $\ell_{u_{b_k - 2}} = L_{\max}^{1/M}$ which implies $\rho_{b_k - 2} \leq \frac{L_{\max}^{1/M} - 2}{L_{\max}^{1/M}} \frac{r_k}{\ell_k}$. (2) If $u_{b_k - 2} \neq u_{b_k - 1}$, which means the algorithm switches to another task at $t = b_k - 1$. Then we have already know that $\rho_{b_k - 1} \leq \frac{L_{\max}^{1/M} - 1}{L_{\max}^{1/M}} \frac{r_k}{\ell_k}$, and similar analysis for $t = b_k - 2$ by setting $b_{u_{b_k - 2}} = b_k - 2$, $\ell_{u_{b_k - 2}} = L_{\max}^{1/M}$, the density is maximized by $\rho_{b_k - 2} \leq (\frac{L_{\max}^{1/M} - 1}{L_{\max}^{1/M}})^2 \frac{r_k}{\ell_k}$. Since $(\frac{L_{\max}^{1/M} - 1}{L_{\max}^{1/M}})^2 > \frac{L_{\max}^{1/M} - 2}{L_{\max}^{1/M}}$, $\rho_{b_k - 2}$ is maximized by $(\frac{L_{\max}^{1/M} - 1}{L_{\max}^{1/M}})^2 \frac{r_k}{\ell_k}$.

We prove the following bound of $\rho_t$ by induction. Suppose for any $i' \leq i$, $\rho_{b_k - i} \leq (\frac{L_{\max}^{1/M} - 1}{L_{\max}^{1/M}})^i \frac{r_k}{\ell_k}$. Then consider time $t = b_k - (i + 1)$, suppose the algorithm processes task $u_t$ at $t = b_k - (i + 1)$ and switches to another task at $t'$ with $t < t' \leq b_k$. Since the maximum density at $t'$ is $(\frac{L_{\max}^{1/M} - 1}{L_{\max}^{1/M}})^{b_k - t'} \frac{r_k}{\ell_k}$. By the selecting rule we have that $\frac{r_{u_t}}{\ell_{u_t} - (t' - b_{u_t})} < (\frac{L_{\max}^{1/M} - 1}{L_{\max}^{1/M}})^{b_k - t'} \frac{r_k}{\ell_k}$, which implies $\frac{r_{u_t}}{\ell_{u_t} - (t - b_{u_t})} < \frac{\ell_{u_t} - (t' - b_{u_t})}{\ell_{u_t} - (t - b_{u_t})} (\frac{L_{\max}^{1/M} - 1}{L_{\max}^{1/M}})^{b_k - t'} \frac{r_k}{\ell_k}$. This is maximized by setting $b_{u_t} = t$, $\ell_{u_t} = L_{\max}^{i/M}$, and we have

$$
\begin{aligned}
\rho_t &\leq \frac{L_{\max}^{1/M} - (t' - t)}{L_{\max}^{1/M}} \left(\frac{L_{\max}^{1/M} - 1}{L_{\max}^{1/M}}\right)^{b_k - t'} \frac{r_k}{\ell_k} \\
&= \prod_{j=1}^{t' - t} \frac{L_{\max}^{1/M} - j}{L_{\max}^{1/M} - j + 1} \left(\frac{L_{\max}^{1/M} - 1}{L_{\max}^{1/M}}\right)^{b_k - t} \frac{r_k}{\ell_k} \\
&\leq \left(\frac{L_{\max}^{1/M} - 1}{L_{\max}^{1/M}}\right)^{t' - t} \left(\frac{L_{\max}^{1/M} - 1}{L_{\max}^{1/M}}\right)^{b_k - t'} \frac{r_k}{\ell_k} \\
&\leq \left(\frac{L_{\max}^{1/M} - 1}{L_{\max}^{1/M}}\right)^{b_k - t} \frac{r_k}{\ell_k} \\
&= \left(\frac{L_{\max}^{1/M} - 1}{L_{\max}^{1/M}}\right)^{i + 1} \frac{r_k}{\ell_k},
\end{aligned}
$$

which is maximized by switching at time $t + 1$. Thus we can conclude that for each time $t \leq b_k$, $\rho_t \leq \left(\frac{L_{\max}^{1/M} - 1}{L_{\max}^{1/M}}\right)^{b_k - t} \frac{r_k}{\ell_k}$. Thus

$$
\sum_{t = b_{k_{pre}} + \ell_{k_{pre}}}^{b_k - 1} \rho_t \leq \sum_{t = b_{k_{pre}} + \ell_{k_{pre}}}^{b_k - 1} \left(\frac{L_{\max}^{1/M} - 1}{L_{\max}^{1/M}}\right)^{b_k - t} \frac{r_k}{\ell_k} \leq L_{\max}^{1/M} \frac{r_k}{\ell_k}.
$$

This implies that

$$
r_k \geq \frac{\ell_k}{L_{\max}^{1/M}} \sum_{t = b_{k_{pre}} + \ell_{k_{pre}}}^{b_k - 1} \rho_t \geq \frac{\ell_k}{L_{\max}^{1/M}} \sum_{k' \in \mathcal{L}_{OPT, k, 1}} r_{k'}.
$$

For optimal tasks in $\mathcal{L}_{OPT, k, 2}$, the maximum total reward is obtained by setting optimal task $k'$ with $b_{k'} = t$, $\ell_{k'} = L_{\max}^{(i-1)/M}$, $r_{k'} = \frac{r_k}{\ell_k - (t - b_k)} \ell_{k'}$ (the maximum reward one can set when no switch happens). Then we have

$$
\sum_{k' \in \mathcal{L}_{OPT, k, 2}} r_{k'} \leq \sum_{t = b_k}^{b_k + \ell_k - 1} \frac{r_k}{\ell_k - (t - b_k)} \leq \ell_k r_k.
$$

This indicates

$$r_k \geq \frac{1}{\log \ell_k} \sum_{k' \in \mathcal{L}_{OPT,k,2}} r_{k'} .$$

For optimal tasks in $\mathcal{L}_{OPT,k,3}$, we know that this set contains at most one task that is completed after $b_k + \ell_k$ (other tasks will be classified to the next task set). This optimal task $k'$ is maximized by setting $b_{k'} = b_k + \ell_k - 1, \ell_{k'} = L_{\max}^{i/M}, r_{k'} = L_{\max}^{1/M} r_k$. And thus we have

$$r_k \geq \frac{1}{L_{\max}^{1/M}} r_{k'} .$$

Thus together we have

$$r_k \geq \frac{\ell_k}{L_{\max}^{1/M}} \sum_{k' \in \mathcal{L}_{OPT,k,1}} r_{k'} ,$$

$$r_k \geq \frac{1}{\ell_k} \sum_{k' \in \mathcal{L}_{OPT,k,2}} r_{k'} ,$$

$$r_k \geq \frac{1}{L_{\max}^{1/M}} \sum_{k' \in \mathcal{L}_{OPT,k,3}} r_{k'} ,$$

which implies

$$r_k \geq \frac{1}{3L_{\max}^{1/M}} \sum_{k' \in \mathcal{L}_{OPT,k,1} \cup \mathcal{L}_{OPT,k,2} \cup \mathcal{L}_{OPT,k,3}} r_{k'}$$

$$= \frac{1}{3L_{\max}^{1/M}} \sum_{k' \in \mathcal{L}_{OPT,k}} r_{k'} .$$

Since there are totally $M$ machines, the sum of rewards in $\mathcal{L}_{OPT}$ needs to time $M$. Summing over all tasks $k \in \mathcal{L}_{ALG}$,

$$\sum_{k \in \mathcal{L}_{ALG}} r_k$$

$$\geq \frac{1}{3ML_{\max}^{1/M}} \sum_{k \in \mathcal{L}_{ALG}} \sum_{k' \in \mathcal{L}_{OPT,k}} r_{k'}$$

$$= \frac{1}{3ML_{\max}^{1/M}} \sum_{k' \in \mathcal{L}_{OPT}} r_{k'} .$$

The algorithm can achieve $1/(3ML_{\max}^{1/M})$ competitive ratio.

$\square$

## C  ANALYSIS FOR S-UCB ALGORITHM

### C.1  PROOF OF LEMMA 5.2

For completed task $k$, conditioned on event $\mathcal{G}_k$, it holds that $\forall k' \in \mathcal{N}_k, \mu_{n(k)}/\ell_k > \mu_{n(k')}/\ell_{k'}$.

We can define the optimal task set $\mathcal{L}_{OPT,k} = \{k' \in \mathcal{L}_{OPT} : b_{k-1} + \ell_{k-1} \leq b_{k'} < b_k + \ell_k\}$ as the set of tasks in $\mathcal{L}_{OPT}$ such that it arrives after $k-1$ is completed and before $k$ is completed. By this split it is easy to verify that the union of $\mathcal{L}_{OPT,k}$ covers all tasks in $\mathcal{L}_{OPT}$.

$$\bigcup_{k \in \mathcal{K}_{ALG}} \mathcal{L}_{OPT,k} = \mathcal{L}_{OPT} .$$

Thus it suffices to analyze the reward relationship between $r_k$ and $\sum_{k' \in \mathcal{L}_{OPT,k}} r_{k'}$ for each task $k \in \mathcal{L}_{ALG}$.

For the set $\mathcal{L}_{OPT,k}$, we further split it into three subsets: $\mathcal{L}_{OPT,k,1} = \{k' \in \mathcal{L}_{OPT,k} : b_{k'} + \ell_{k'} \leq b_k\}$ is the set of optimal tasks completed before $k$ arrives; $\mathcal{L}_{OPT,k,2} = \{k' \in \mathcal{L}_{OPT,k} : b_{k'} \geq b_k, b_{k'} + \ell_{k'} \leq b_k + \ell_k\}$ is the set of optimal tasks arrives after $k$ arrives and completed before $k$ is completed; $\mathcal{L}_{OPT,k,3} = \{k' \in \mathcal{L}_{OPT,k} : b_{k'} + \ell_{k'} > b_k + \ell_k\}$ is the set of optimal tasks completed after $k$ is completed. Since tasks in $\mathcal{L}_{OPT}$ are disjoint, it is easy to check that

$$\mathcal{L}_{OPT,k,1} \cup \mathcal{L}_{OPT,k,2} \cup \mathcal{L}_{OPT,k,3} = \mathcal{L}_{OPT,k}.$$

We first upper bound the rewards of tasks in $\mathcal{L}_{OPT,k,1}$ for S-UCB algorithm conditioned on $\mathcal{G}_k$. Since between time $b_{k-1} + \ell_{k-1}$ and $b_k$ the algorithm completes no task, this indicates that the algorithm keeps processing some tasks at those times but switches to other tasks before it is completed, and in the end, the algorithm switches to task $k$ at $t = b_k$ and completes it. Denote $u_t$ as the processing task by algorithm at time $t$ and $\rho_t = r_{u_t}/(\ell_{u_t} - (t - b_{u_t}))$ as the remaining density of that processing task. By the algorithm's selecting rule, $\rho_t$ is the maximum density of all arriving tasks at time $t$. Thus the total rewards in $\mathcal{L}_{OPT,k,1}$ is maximized by

$$\sum_{k' \in \mathcal{L}_{OPT,k,1}} r_{k'} \leq \sum_{t=b_{k-1}+\ell_{k-1}}^{b_k-1} \rho_t.$$

Then it suffices to upper bound $\rho_t$ at each time $t$. Since the algorithm does not complete any task at $t \in [b_{k-1} + \ell_{k-1}, b_k]$, the density $\rho_t$ is increasing with time $t$ and $k$ has the highest density by $\mathcal{G}_k$.

We first consider the density $\rho_t$ at $t = b_k - 1$. Recall that $u_{b_k-1}$ is the processing task at time $t = b_k - 1$ and $\rho_t = r_{u_t}/(\ell_{u_t} - (t - b_{u_t}))$. Since the algorithm switches to $k$ at time $b_k$, it indicates that $r_k/\ell_k > r_{u_t}/(\ell_{u_t} - (t - b_{u_t}) - 1)$. Thus $\rho_t = r_{u_t}/(\ell_{u_t} - (t - b_{u_t})) < \frac{r_k}{\ell_k} \frac{\ell_{u_t} - (t-b_{u_t}) - 1}{\ell_{u_t} - (t-b_{u_t})}$, which is maximized by setting $\ell_{u_t} = L_{\max}^{1/M}, b_{u_t} = t$, and thus $\rho_t \leq \frac{L_{\max}^{1/M} - 1}{L_{\max}^{1/M}} \frac{r_k}{\ell_k}$ at time $t = b_k - 1$.

Then we turn to consider the density $\rho_t$ at $t = b_k - 2$. (1) If $u_{b_k-2} = u_{b_k-1}$, which means the algorithm does not switch at $t = b_k - 1$, and switches to task $k$ at $t = b_k$. Thus we have $\frac{r_k}{\ell_k} > \frac{r_{u_{b_k-2}}}{\ell_{u_{b_k-2}} - (b_k - b_{u_{b_k-2}})}$. This is maximized by setting $b_{u_{b_k-2}} = b_k - 2, \ell_{u_{b_k-2}} = L_{\max}^{1/M}$ which implies $\rho_{b_k-2} \leq \frac{L_{\max}^{1/M} - 2}{L_{\max}^{1/M}} \frac{r_k}{\ell_k}$. (2) If $u_{b_k-2} \neq u_{b_k-1}$, which means the algorithm switches to another task at $t = b_k - 1$. Then we have already know that $\rho_{b_k-1} \leq \frac{L_{\max}^{1/M} - 1}{L_{\max}^{1/M}} \frac{r_k}{\ell_k}$, and similar analysis for $t = b_k - 2$ by setting $b_{u_{b_k-2}} = b_k - 2, \ell_{u_{b_k-2}} = L_{\max}^{1/M}$, the density is maximized by $\rho_{b_k-2} \leq (\frac{L_{\max}^{1/M} - 1}{L_{\max}^{1/M}})^2 \frac{r_k}{\ell_k}$. Since $(\frac{L_{\max}^{1/M} - 1}{L_{\max}^{1/M}})^2 > \frac{L_{\max}^{1/M} - 2}{L_{\max}^{1/M}}$, $\rho_{b_k-2}$ is maximized by $(\frac{L_{\max}^{1/M} - 1}{L_{\max}^{1/M}})^2 \frac{r_k}{\ell_k}$.

We prove the following bound of $\rho_t$ by induction. Suppose for any $i' \leq i$, $\rho_{b_k-i} \leq (\frac{L_{\max}^{1/M} - 1}{L_{\max}^{1/M}})^i \frac{r_k}{\ell_k}$. Then consider time $t = b_k - (i+1)$, suppose the algorithm processes task $u_t$ at $t = b_k - (i+1)$ and switches to another task at $t'$ with $t < t' \leq b_k$. Since the maximum density at $t'$ is $(\frac{L_{\max}^{1/M} - 1}{L_{\max}^{1/M}})^{b_k-t'} \frac{r_k}{\ell_k}$. By the selecting rule we have that $\frac{r_{u_t}}{\ell_{u_t} - (t' - b_{u_t})} < (\frac{L_{\max}^{1/M} - 1}{L_{\max}^{1/M}})^{b_k-t'} \frac{r_k}{\ell_k}$, which implies $\frac{r_{u_t}}{\ell_{u_t} - (t - b_{u_t})} <$

$\frac{\ell_{u_t} - (t' - b_{u_t})}{\ell_{u_t} - (t - b_{u_t})} \left( \frac{L_{\max}^{1/M} - 1}{L_{\max}^{1/M}} \right)^{b_k - t'} \frac{r_k}{\ell_k}$. This is maximized by setting $b_{u_t} = t, \ell_{u_t} = L_{\max}^{i/M}$, and we have

$$\rho_t \leq \frac{L_{\max}^{1/M} - (t' - t)}{L_{\max}^{1/M}} \left( \frac{L_{\max}^{1/M} - 1}{L_{\max}^{1/M}} \right)^{b_k - t'} \frac{r_k}{\ell_k}$$

$$= \prod_{j=1}^{t'-t} \frac{L_{\max}^{1/M} - j}{L_{\max}^{1/M} - j + 1} \left( \frac{L_{\max}^{1/M} - 1}{L_{\max}^{1/M}} \right)^{b_k - t} \frac{r_k}{\ell_k}$$

$$\leq \left( \frac{L_{\max}^{1/M} - 1}{L_{\max}^{1/M}} \right)^{t'-t} \left( \frac{L_{\max}^{1/M} - 1}{L_{\max}^{1/M}} \right)^{b_k - t'} \frac{r_k}{\ell_k}$$

$$\leq \left( \frac{L_{\max}^{1/M} - 1}{L_{\max}^{1/M}} \right)^{b_k - t} \frac{r_k}{\ell_k}$$

$$= \left( \frac{L_{\max}^{1/M} - 1}{L_{\max}^{1/M}} \right)^{i+1} \frac{r_k}{\ell_k},$$

which is maximized by switching at time $t + 1$. Thus we can conclude that for each time $t \leq b_k$, $\rho_t \leq \left( \frac{L_{\max}^{1/M} - 1}{L_{\max}^{1/M}} \right)^{b_k - t} \frac{r_k}{\ell_k}$. Thus

$$\sum_{t=b_{k_{pre}} + \ell_{k_{pre}}}^{b_k - 1} \rho_t \leq \sum_{t=b_{k_{pre}} + \ell_{k_{pre}}}^{b_k - 1} \left( \frac{L_{\max}^{1/M} - 1}{L_{\max}^{1/M}} \right)^{b_k - t} \frac{r_k}{\ell_k} \leq L_{\max}^{1/M} \frac{r_k}{\ell_k}.$$

This implies that

$$r_k \geq \frac{\ell_k}{L_{\max}^{1/M}} \sum_{t=b_{k_{pre}} + \ell_{k_{pre}}}^{b_k - 1} \rho_t \geq \frac{\ell_k}{L_{\max}^{1/M}} \sum_{k' \in \mathcal{L}_{OPT,k,1}} r_{k'}.$$

For optimal tasks in $\mathcal{L}_{OPT,k,2}$, the maximum total reward is obtained by setting optimal task $k'$ with $b_{k'} = t, \ell_{k'} = L_{\max}^{(i-1)/M}, r_{k'} = \frac{r_k}{\ell_k - (t - b_k)} \ell_{k'}$ (the maximum reward one can set when no switch happens). Then we have

$$\sum_{k' \in \mathcal{L}_{OPT,k,2}} r_{k'} \leq \sum_{t=b_k}^{b_k + \ell_k - 1} \frac{r_k}{\ell_k - (t - b_k)} \leq \ell_k r_k.$$

This indicates

$$r_k \geq \frac{1}{\log \ell_k} \sum_{k' \in \mathcal{L}_{OPT,k,2}} r_{k'}.$$

For optimal tasks in $\mathcal{L}_{OPT,k,3}$, we know that this set contains at most one task that is completed after $b_k + \ell_k$ (other tasks will be classified to the next task set). This optimal task $k'$ is maximized by setting $b_{k'} = b_k + \ell_k - 1, \ell_{k'} = L_{\max}^{i/M}, r_{k'} = L_{\max}^{1/M} r_k$. And thus we have

$$r_k \geq \frac{1}{L_{\max}^{1/M}} r_{k'}.$$

Thus together we have

$$r_k \geq \frac{\ell_k}{L_{\max}^{1/M}} \sum_{k' \in \mathcal{L}_{OPT,k,1}} r_{k'},$$

$$r_k \geq \frac{1}{\ell_k} \sum_{k' \in \mathcal{L}_{OPT,k,2}} r_{k'},$$

$$r_k \geq \frac{1}{L_{\max}^{1/M}} \sum_{k' \in \mathcal{L}_{OPT,k,3}} r_{k'},$$

which implies

$$r_k \geq \frac{1}{3L_{\max}^{1/M}} \sum_{k' \in \mathcal{L}_{OPT,k,1} \cup \mathcal{L}_{OPT,k,2} \cup \mathcal{L}_{OPT,k,3}} r_{k'}$$

$$= \frac{1}{3L_{\max}^{1/M}} \sum_{k' \in \mathcal{L}_{OPT,k}} r_{k'}.$$

Since there are totally $M$ machines, the sum of rewards in $\mathcal{L}_{OPT}$ needs to time $M$. Summing over all tasks $k \in \mathcal{L}_{ALG}$,

$$\sum_{k \in \mathcal{L}_{ALG}} r_k$$

$$\geq \frac{1}{3ML_{\max}^{1/M}} \sum_{k \in \mathcal{L}_{ALG}} \sum_{k' \in \mathcal{L}_{OPT,k}} r_{k'}$$

$$= \frac{1}{3ML_{\max}^{1/M}} \sum_{k' \in \mathcal{L}_{OPT}} r_{k'}.$$

Thus the S-UCB algorithm can achieve $1/(3ML_{\max}^{1/M})$ competitive ratio conditioned on $\mathcal{G}_k$, implying 0 approximate regret.

## C.2  PROOF OF LEMMA 5.3

The regret caused by exploring the UCB index in S-UCB algorithm is bounded by

$$\mathbb{E}\left[\frac{1}{3ML_{\max}^{1/M}} \sum_{i=1}^{M} \sum_{k \in \mathcal{K}_{\mathrm{ALG}}^i} \mathbb{1}\{\neg\mathcal{G}_k\}\big(\sum_{k' \in \mathcal{N}_k} r_{k'} - r_k\big)\right] \leq \sum_{n \in [N]} O\left(\frac{N L_{\max} \log T}{\Delta_n}\right) \leq O\left(\frac{N^2 L_{\max} \log T}{\Delta}\right),$$

*Proof.* We can decompose the regret term by its type:

$$\mathbb{E}\left[\frac{1}{3ML_{\max}^{1/M}} \sum_{i=1}^{M} \sum_{k \in \mathcal{K}_{\mathrm{ALG}}^i} \mathbb{1}\{\neg\mathcal{G}_k\}\big(\sum_{k' \in \mathcal{N}_k} r_{k'} - r_k\big)\right]$$

$$\leq \mathbb{E}\left[\frac{1}{3ML_{\max}^{1/M}} \sum_{i=1}^{M} \sum_{n=1}^{N} \sum_{k \in \mathcal{K}_{\mathrm{ALG}}^i, n(k)=n} \mathbb{1}\{\neg\mathcal{G}_k\}\big(\sum_{k' \in \mathcal{N}_k} r_{k'} - r_k\big)\right],$$

We define the failure event $\mathcal{F} = \{\exists i \in [N], |\hat{\mu}_i(t) - \mu_i| > \sqrt{\frac{6\log(T)}{T_i(t)}}\}$. The probability of the failure event is bounded by the following lemma.

**Lemma C.1.** *Let $\mathcal{F} = \{\exists i \in [N], |\hat{\mu}_i(t) - \mu_i| > \sqrt{\frac{6\log(T)}{T_i(t)}}\}$ be the bad event that some tasks' rewards are not estimated well at time $t$, we have that*

$$\mathbb{P}\left(\mathcal{F}\right) \leq 2N/T.$$

*Proof.*

$$\mathbb{P}\left(\mathcal{F}\right) = \mathbb{P}\left(\exists 1 \le t \le T, i \in [N] : |\hat{\mu}_i(t) - \mu_i| > \sqrt{\frac{6\log T}{T_i(t)}}\right)$$

$$\le \sum_{t=1}^{T} \sum_{i \in [N]} \mathbb{P}\left(|\hat{\mu}_i(t) - \mu_i| > \sqrt{\frac{6\log T}{T_i(t)}}\right)$$

$$\le \sum_{t=1}^{T} \sum_{i \in [N]} \sum_{s=1}^{t} \mathbb{P}\left(T_i(t) = s, |\hat{\mu}_i(t) - \mu_i| > \sqrt{\frac{6\log T}{s}}\right)$$

$$\le \sum_{t=1}^{T} \sum_{i \in [N]} t \cdot 2\exp(-3\log T)$$

$$\le 2N/T,$$

where the second last inequality is derived from the Hoeffding inequality. $\square$

Then under event $\neg\mathcal{F}$, the empirical mean $\hat{\mu}_n$ is upper bounded by $\mu_n + \sqrt{\frac{6\log T}{T_n(t)}}$. Thus the UCB index is bounded by $\mu_n + 2\sqrt{\frac{6\log T}{T_n(t)}}$.

Given task $k$ and $k'$ and S-UCB completes task $k$, suppose $\mu_{n(k)}/\ell_k < \mu_{n(k')}/\ell_{k'}$, we need to bound the number of times $\text{UCB}_{n(k)}/\ell_k > \text{UCB}_{n(k')}/\ell_{k'}$, which is the wrong estimation and incurs regret. Denote $\Delta_{k,k'} = \mu_{n(k')}/\ell_{k'} - \mu_{n(k)}/\ell_k$. By the upper bound of UCB index, we have that when $T_{n(k)} > 24\log T/\Delta_{k,k'}^2$, it holds that

$$\text{UCB}_{n(k)}/\ell_k \le \mu_{n(k)}/\ell_k + 2\sqrt{\frac{6\log T}{T_n(t)}}$$

$$\le \mu_{n(k)}/\ell_k + \Delta_{k,k'}$$

$$\le \mu_{n(k')}/\ell_{k'}.$$

This indicates the the number of selections of task $k$ is bounded by $24\log T/\Delta_{k,k'}^2$. Additionally, each time it completes $k$, the additional reward for $k'$ beyond competitive ratio is upper bounded by $\Delta_{k,k'}\ell_{k'}$. Thus the regret is bounded by

$$\mathbb{E}\left[\frac{1}{3ML_{\max}^{1/M}} \sum_{i=1}^{M} \sum_{k \in \mathcal{K}_{\text{ALG}}^i} \mathbb{1}\{\neg\mathcal{G}_k\} \left(\sum_{k' \in \mathcal{N}_k} r_{k'} - r_k\right)\right]$$

$$\le \mathbb{E}\left[\frac{1}{3ML_{\max}^{1/M}} \sum_{i=1}^{M} \sum_{n=1}^{N} \sum_{k \in \mathcal{K}_{\text{ALG}}^i, n(k)=n} \mathbb{1}\{\neg\mathcal{G}_k\} \left(\sum_{k' \in \mathcal{N}_k} r_{k'} - r_k\right) | \neg\mathcal{F}\right] + T\mathbb{P}\left(\mathcal{F}\right)$$

$$\le \mathbb{E}\left[\frac{1}{3ML_{\max}^{1/M}} \sum_{i=1}^{M} \sum_{n=1}^{N} \sum_{n' \ne n} \Delta_{n,n'} L_{\max} \frac{24\log T}{\Delta_{n,n'}^2} | \neg\mathcal{F}\right] + 2N$$

$$\le \sum_{n=1}^{N} \frac{24NL_{\max}\log T}{\Delta_n} + 2N$$

$$\le \frac{24N^2 L_{\max}\log T}{\Delta} + 2N,$$

where $\Delta_{n,n'}$ is the minimum density gap between task $n$ and $n'$. $\square$

# D    PROOF OF THEOREM 5.5

**Theorem D.1.** *Given instance $I$, following the S-UCB algorithm (Algorithm 2), the scheduler can achieve the expected competitive ratio with*

$$\mathbb{E}\left[R\right] \geq \frac{1}{3ML_{\max}^{1/M} + N^2 L_{\max}^{1/M} \log T/(R_{ALG}(I)\Delta)} ,$$

*where $R_{ALG}(I)$ is the total completion rewards obtained by S-UCB, and expectation is taken over the randomness of reward distribution.*

*Proof.*

$$
\begin{aligned}
\mathbb{E}\left[R\right] =& \mathbb{E}\left[\frac{R_{\mathrm{ALG}}}{R_{\mathrm{OPT}}}\right] \\
=& \mathbb{E}\left[\frac{\sum_{k\in\mathcal{K}_{\mathrm{ALG}}} r_k}{\sum_{k\in\mathcal{K}_{\mathrm{OPT}}} r_k}\right] \\
=& \mathbb{E}\left[\frac{\sum_{k\in\mathcal{K}_{\mathrm{ALG}}} \mathbb{1}\{\mathcal{G}_k\} r_k + \sum_{k\in\mathcal{K}_{\mathrm{ALG}}} \mathbb{1}\{\neg\mathcal{G}_k\} r_k}{\sum_{k\in\mathcal{K}_{\mathrm{ALG}}} \mathbb{1}\{\mathcal{G}_k\} \sum_{k'\in\mathcal{N}_k} r_{k'} + \sum_{k\in\mathcal{K}_{\mathrm{ALG}}} \mathbb{1}\{\neg\mathcal{G}_k\} \sum_{k'\in\mathcal{N}_k} r_{k'}}\right] \\
\geq& \mathbb{E}\left[\frac{\sum_{k\in\mathcal{K}_{\mathrm{ALG}}} \mathbb{1}\{\mathcal{G}_k\} r_k + \sum_{k\in\mathcal{K}_{\mathrm{ALG}}} \mathbb{1}\{\neg\mathcal{G}_k\} r_k}{3ML_{\max}^{1/M} \sum_{k\in\mathcal{K}_{\mathrm{ALG}}} \mathbb{1}\{\mathcal{G}_k\} r_k + 3ML_{\max}^{1/M} \sum_{k\in\mathcal{K}_{\mathrm{ALG}}} \mathbb{1}\{\neg\mathcal{G}_k\} r_k + 24N^2 L_{\max} \log T/\Delta}\right] \\
=& \mathbb{E}\left[\frac{1}{3ML_{\max}^{1/M} + 24N^2 L_{\max} \log T/\Delta R_{\mathrm{ALG}}(I)}\right] .
\end{aligned}
$$

$\square$

# E    EXPLORE THEN SCHEDULING ALGORITHM

The key idea to balance exploration and exploitation is that the algorithm always performs MRDF algorithm (Algorithm 1) when it is confident to determine the maximum remaining density of the coming tasks. This confidence is computed by constructing an upper confidence bound and a lower confidence bound for each task's mean reward estimation. And the algorithm will enter the exploration stage if it is still not confident to perform the MRDF algorithm. This motivates us to design the following Explore then Scheduling algorithm (Algorithm 3).

The algorithm first initializes $\mathrm{UCB}_n, \mathrm{LCB}_n, T_n$ for each type $n \in [N]$, where $\mathrm{UCB}_n, \mathrm{LCB}_n$ are the upper confidence bound (UCB) index and lower confidence bound (LCB) index for type $n$, respectively. $T_n$ is the number of times task $n$ is completed. Denote $\mathcal{M}$ as the task set the machine is processing and it is empty at the beginning. Define FLAG as the sign of the current algorithm's stage, and it is set as "Exploit" stage at first.

At time $t$, if the current processing task $k \in \mathcal{M}$ is completed, i.e., $b_k + \ell_k = t$, then the algorithm will update the corresponding $\mathrm{UCB}_n, \mathrm{LCB}_n, T_n$ for type $n = n(k)$ (Line 4). Specifically, $T_n = T_n + 1$, the corresponding upper confidence bound $\mathrm{UCB}_n = \hat{\mu}_n + \sqrt{\frac{6\log T}{T_n}}$, and lower confidence bound $\mathrm{LCB}_n = \hat{\mu}_n - \sqrt{\frac{6\log T}{T_n}}$, where $\hat{\mu}_n$ is the empirical reward mean of task $n$. After that, the algorithm resets the FLAG to the Exploit stage (Lines 5 - 9).

Then tasks set $\mathcal{L}_t$ comes and the algorithm observes $b_k, \ell_k$ and $n(k)$ for each coming task $k \in \mathcal{L}_t$ (Line 10). If the current algorithm's stage FLAG = Explore, then it means the algorithm needs to explore the current processing task $k \in \mathcal{M}$, and thus the algorithm will do nothing at time $t$ (Line 11 - 12). When current stage FLAG = Exploit, similarly with MRDF (Algorithm 1), the algorithm first constructs a pseudo task $k'$ for $k \in \mathcal{M}$ with $b_{k'} = t$, $\ell_{k'} = \ell_k - (t - b_k)$, $n(k') = n(k)$, and add this pseudo task $k'$ to $\mathcal{L}_t$ (Lines 15 - 18.) Then the algorithm checks if the maximum $r/\ell$ can be distinguished, i.e., $\exists k, \forall k' \neq k, \frac{\mathrm{LCB}_{n(k)}}{\ell_k} > \frac{\mathrm{UCB}_{n(k')}}{\ell_{k'}}$. This indicates that task $k$ has a maximum

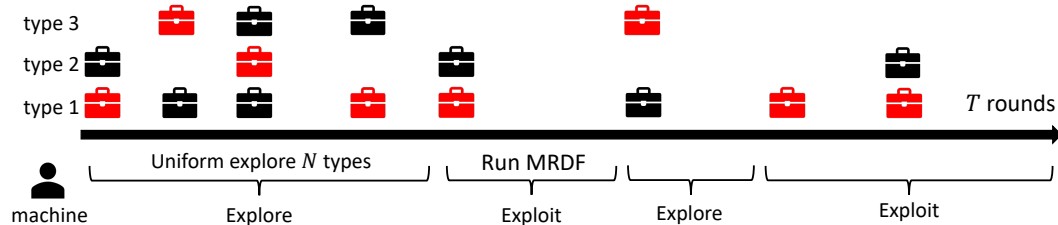

Figure 4: Illustration of Explore then Scheduling algorithm. There are $T$ rounds and 3 types of tasks, in each round some types of tasks come. The red color denotes the processed task. If the algorithm is not confident which task has the maximum density, it explores these 3 types uniformly (in the "Explore" stage). Otherwise, the algorithm performs the same as MRDF algorithm (in the "Exploit" stage).

$r/\ell$ with high probability, and thus the algorithm runs the same process as the online algorithm with the known mean reward (Lines 19 - 21). Otherwise there exists an overlap between the largest LCB index and another UCB index, the algorithm will turn to the Explore stage and select the task with minimum $T_{n(k)}$ to guarantee the uniform exploration over all types (Lines 22 - 26).

---

**Algorithm 3** Explore then Scheduling

1: Initialize: $\text{UCB}_n = \infty, \text{LCB}_n = -\infty, T_n = 0, \forall n \in [N], \text{FLAG} = \text{Exploit}, \mathcal{M} = \emptyset$;
2: **for** $t = 1, 2, \cdots$ **do**
3:    **if** $\mathcal{M} \neq \emptyset$ and $b_k + \ell_k = t, k \in \mathcal{M}$ **then**
4:       Receive reward $r_k$, update $\text{UCB}_{n(k)}, \text{LCB}_{n(k)}$, and $T_{n(k)}$;
5:       $\mathcal{M} = \emptyset$;
6:       **if** FLAG = Explore **then**
7:          FLAG = Exploit;
8:       **end if**
9:    **end if**
10:    Receive coming tasks set $\mathcal{L}_t = \{k \mid b_k = t\}$, observe $b_k, \ell_k, n(k)$ for each $k \in \mathcal{L}_t$;
11:    **if** FLAG = Explore **then**
12:       Continue processing current task $k \in \mathcal{M}$;
13:    **end if**
14:    **if** FLAG = Exploit **then**
15:       **if** $\mathcal{M} \neq \emptyset$ **then**
16:          For $k \in \mathcal{M}$, construct pseudo task $k'$ with $b_{k'} = t, \ell_{k'} = \ell_k - (t - b_k), n(k') = n(k)$;
17:          $\mathcal{L}_t = \mathcal{L}_t \cup \{k'\}$;
18:       **end if**
19:       Find task $k \in \arg\max_{k' \in \mathcal{L}_t} \frac{\text{LCB}_{n(k')}}{\ell_{k'}}$;
20:       **if** $\frac{\text{LCB}_{n(k)}}{\ell_k} > \max_{k' \in \mathcal{L}_t \setminus \{k\}} \frac{\text{UCB}_{n(k')}}{\ell_{k'}}$ **then**
21:          $\mathcal{M} = \{k\}$;
22:       **else**
23:          FLAG = Explore;
24:          Find $k' \in \arg\min_{k'' \in \mathcal{L}_t} T_{n(k'')}$;
25:          $\mathcal{M} = \{k'\}$;
26:       **end if**
27:    **end if**
28: **end for**

---

**Theorem E.1.** *When the learner follows the Online then Scheduling algorithm (Algorithm 3), the learner can obtain the $\frac{1}{3ML_{\max}^{1/M}}$-approximate regret bounded by*

$$\text{Reg}(T) \leq \frac{24NL_{\max}\log T}{\Delta^2} + 2N,$$

where $\Delta = \min_{n \neq n', \ell, \ell'} |\frac{\mu_n}{\ell} - \frac{\mu_{n'}}{\ell'}|$ is the minimum density gap of two tasks.

Denote $r(t)$ as the reward the algorithm gets at time $t$, and $r_{\mathrm{OPT}}(t)$ is the reward the compared optimal scheduling strategy OPT gets at time $t$. Then the regret can be rewritten as

$$
\mathrm{Reg}(T)
$$

$$
= \mathbb{E}\left[\frac{1}{3L_{\max}} \sum_{t=1}^{T} r_{\mathrm{OPT}}(t) - \sum_{t=1}^{T} r(t)\right]
$$

$$
= \mathbb{E}\left[\sum_{t=1}^{T} \mathbb{1}\{\mathrm{FLAG}_t = \mathrm{Exploit}\}\,(\frac{1}{3L_{\max}} r_{\mathrm{OPT}}(t) - r(t))\right]
$$

$$
+ \mathbb{E}\left[\sum_{t=1}^{T} \mathbb{1}\{\mathrm{FLAG}_t = \mathrm{Explore}\}\,(\frac{1}{3L_{\max}} r_{\mathrm{OPT}}(t) - r(t))\right],
$$

where $\mathrm{FLAG}_t$ is the FLAG at time $t$ performed by Algorithm 3. The regret is decomposed by two terms: the first is the regret caused by the exploitation stage, and the second term is the regret caused by exploration stage.

To bound the regret caused by the exploitation stage, the following two lemmas state that with high probability, the algorithm performs the same as the MRDF algorithm (Algorithm 3) with known mean reward, which attains $1/(3L_{\max})$ competitive ratio. Thus the regret can be bounded by a constant independent of $T$.

**Lemma E.2.** *Let* $\mathcal{F} = \{\exists i \in [N], |\hat{\mu}_i(t) - \mu_i| > \sqrt{\frac{6\log(T)}{T_i(t)}}\}$ *be the bad event that some tasks' rewards are not estimated well at time $t$, we have that*

$$
\mathbb{P}\left(\mathcal{F}\right) \leq 2N/T.
$$

*Proof.*

$$
\mathbb{P}\left(\mathcal{F}\right) = \mathbb{P}\left(\exists 1 \leq t \leq T, i \in [N] : |\hat{\mu}_i(t) - \mu_i| > \sqrt{\frac{6\log T}{T_i(t)}}\right)
$$

$$
\leq \sum_{t=1}^{T} \sum_{i \in [N]} \mathbb{P}\left(|\hat{\mu}_i(t) - \mu_i| > \sqrt{\frac{6\log T}{T_i(t)}}\right)
$$

$$
\leq \sum_{t=1}^{T} \sum_{i \in [N]} \sum_{s=1}^{t} \mathbb{P}\left(T_i(t) = s, |\hat{\mu}_i(t) - \mu_i| > \sqrt{\frac{6\log T}{s}}\right)
$$

$$
\leq \sum_{t=1}^{T} \sum_{i \in [N]} t \cdot 2\exp(-3\log T)
$$

$$
\leq 2N/T,
$$

where the second last inequality is derived from the Hoeffding inequality. $\qquad\square$

**Lemma E.3.** *When* $\exists k, \forall k' \neq k, \frac{LCB_{n(k)}(t)}{\ell_k} > \frac{UCB_{n(k')}(t)}{\ell_{k'}}$, *under event* $\neg\mathcal{F}$, *the algorithm proceeds the same as the online algorithm with known mean reward* $\mu_n$.

*Proof.* When $\exists k, \forall k' \neq k, \frac{LCB_{n(k)}(t)}{\ell_k} > \frac{UCB_{n(k')}(t)}{\ell_{k'}}$, under event $\neg\mathcal{F}$, it indicates that $\exists k, \forall k' \neq k, \frac{\mu_{n(k)}(t)}{\ell_k} > \frac{\mu_{n(k')}(t)}{\ell_{k'}}$. Thus the task with the maximum remaining density is exactly $k$, and the algorithm proceeds the same as MRDF. $\qquad\square$

This indicates that with high probability the algorithm achieves the competitive ratio larger than $1/(3L_{\max})$, and the regret is caused by the low probability of event $\mathcal{F}$.

$$\mathbb{E}\left[\sum_{t=1}^{T} \mathbb{1}\{\text{FLAG}_t = \text{Exploit}\}\,(\frac{1}{3L_{\max}}r_{\text{OPT}}(t) - r(t))\right]$$
$$\leq T\mathbb{P}(\mathcal{F}) \leq 2N\,.$$

Then we turn to bound the second term, which is the regret caused by exploration stage. This term is upper bounded by the maximum reward per task times the number of explorations for each type of task. By algorithm design we have that the learner always explores the task with the minimum number of selections (Line 24 in Algorithm 3). This guarantees that all types of tasks are explored in a round-robin manner. The following lemma upper bounds the exploration times for each task.

**Lemma E.4.** *The number of times the algorithm is exploring is bounded by* $24NL_{\max}\log T/\Delta^2$.

*Proof.* Given two types $n, n'$, we have that conditioned on $\neg\mathcal{F}$, when $T_n(t) \geq 24\log T/\Delta^2$ and $T_{n'}(t) \geq 24\log T/\Delta^2$, then $\forall \ell, \ell'$, $\frac{\text{LCB}_n(t)}{\ell} > \frac{\text{UCB}_{n'}(t)}{\ell'}$ or $\frac{\text{UCB}_n(t)}{\ell} < \frac{\text{LCB}_{n'}(t)}{\ell'}$. By the algorithm design, the learner will always explore the task with the minimum $T_n(t)$, which means each task will reach $\log T/\Delta^2$ uniformly. To explore one task, the leaner will cost at most $L_{\max}$ rounds. Thus the number of times the algorithm is exploring is bounded by $24NL_{\max}\log T/\Delta^2$. $\square$

This indicates that

$$\mathbb{E}\left[\sum_{t=1}^{T} \mathbb{1}\{\text{FLAG}_t = \text{Explore}\}\,(\frac{1}{3L_{\max}}r_{\text{OPT}}(t) - r(t))\right]$$
$$\leq \frac{24NL_{\max}\log T}{\Delta^2}\,.$$

Then summing over these two terms, theorem is derived.

# F    USE OF LARGE LANGUAGE MODEL

This paper did not use LLMs to assist in writing or for other purposes.

