# OpenReview forum: "Bandit Learning for Online Scheduling with Immediate Decision"
_ICLR.cc/2026/Conference — Submitted to ICLR 2026_

### Official Review · Reviewer_DdzM · 2025-10-30

**Soundness:** 3
**Presentation:** 2
**Contribution:** 3
**Rating:** 6
**Confidence:** 2

**Summary:**

This paper studied the online scheduling with immediate decision problem where the tasks arrive streaming and machines need to decide to complete the task or not immediately without a second chance. For this problem, the authors consider two distinct settings, i.e., with or without knowing the reward. For the two cases, algorithms with theoretical guarantees are proposed.

**Strengths:**

1.	This paper proposes a new and interesting research problem.

2.	Strong theoretical results with approximation bounds are derived.

**Weaknesses:**

1.	The paper mentions several real-world applications as instances of the problem. However, the underlying relevance and motivation of this problem are not strong for me. Please further elaborate for clarity.

2.	In the setting in Section 3, the problem considers multiple machines. However, in the description, only one machine is considered. I wonder if a task is discarded by machine A, if it will be considered by another machine B? Do those machines follow the same strategies on decision by sharing information, or do they make independent decisions? If those task rewards follow the same distribution? Please clarify.

3.	Several minor issues: i) this paper is math-heavy, while some notations are not explained before use; ii) the paper is not well formatted, e.g., some formulas are out of scope.

**Questions:**

Please refer to my weakness comments.

---

> ### Author Response · Authors · 2025-11-22
>
> We sincerely thank the reviewer for your thoughtful feedback and for raising these important points. Below, we provide a point-by-point response to the weaknesses identified in the review.
>
> ### 1. Relevance and Motivation of the Problem
>
> We apologize for any lack of clarity regarding the practical motivation of our problem. The online scheduling setting with immediate decisions and preemption is indeed motivated by several real-world applications where tasks must be processed immediately upon arrival or be permanently lost. For instance:
>
> IoT Data Streaming: In sensor networks, data packets arrive continuously and must be processed in real-time for timely decision-making (e.g., environmental monitoring or industrial automation). If not processed immediately, the data becomes obsolete due to rapid generation of new data, and storing all data is infeasible due to resource constraints.
>
> Cloud Computing Resource Allocation: In serverless computing or real-time cloud platforms, function invocations triggered by events (e.g., API requests) require immediate resource allocation. If resources are not available instantly, the invocation fails, leading to service degradation.
>
> Financial Data Processing: In high-frequency trading, market data feeds must be processed within microseconds to execute trades. Delayed processing renders the information worthless, as market conditions change rapidly.
>
> In these applications, the key challenge is the irrevocability of decisions, where tasks cannot be buffered or revisited once discarded. Our work aims to provide a formal model and algorithms for such settings, which are underrepresented in existing literature focused on buffered systems. We will revise the introduction to elaborate on these motivations more clearly, emphasizing the practical necessity of immediate decision-making.
>
> ### 2. Clarification on Multiple Machines and Task Handling
>
> We thank the reviewer for pointing out the confusion regarding the multi-machine setting. In our model, we consider a centralized setting, where there exists a scheduler that assigns tasks to each machine. Thus machines follow a union strategy by a centralized scheduler. For the reward of tasks, the reward distribution is i.i.d for a certain type of task. And the reward distribution is different for different types.
>
> ### 3. Minor Issues: Notation and Formatting
>
> We acknowledge that the paper is math-heavy and that some notations were not adequately explained before use. To address this:
>
> We will add a notation table in the appendix or early in the paper to define all symbols and terms used. Additionally, we will ensure that each notation is introduced clearly upon first use in the text.

---

### Official Review · Reviewer_NiFr · 2025-10-31

**Soundness:** 3
**Presentation:** 3
**Contribution:** 3
**Rating:** 4
**Confidence:** 3

**Summary:**

This paper studies an online scheduling problem with immediate decision-making and uncertain rewards, motivated by applications like IoT and cloud computing. The authors first analyze the known-reward setting, establish a worst-case competitive ratio, and propose a near-optimal algorithm (MRDF). Then, for the unknown reward case, they introduce a bandit-based algorithm (S-UCB) that achieves $O(\log T)$ regret and asymptotically matches the competitive ratio of MRDF. Experiments are provided to validate the theoretical results.

**Strengths:**

1. The problem setup is clearly described. The paper is well-structured and easy to follow.

2. The analysis of the deterministic MRDF algorithm is mathematically clean, and the idea of partitioning machines by task-length ranges is intuitively interesting.

3. The analysis in Section 4 is the strongest part of the paper. Deriving the worst-case competitive ratio (Theorem 4.1) and then proposing an algorithm (MRDF) with a matching bound (Theorem 4.2) is a nice, complete theoretical result.

**Weaknesses:**

1. The main theoretical contributions feel incremental and can be viewed as extensions of prior work in bandit scheduling. The main technical novelty lies in the analysis of the preemptable property of each machine. The MRDF algorithm is essentially a length-partitioned greedy heuristic that heavily builds on classical scheduling principles such as maximum-density-first, while the S-UCB algorithm is just a direct plug-in of the classic UCB index into MRDF.  The preemption mechanism in the bandit scheduling is interesting but not fundamentally new.

2. I think the problem is also related to admission control with reusable resources (e.g., [1, 2]), and the authors should compare their work with this line of literature.

3. The regret analysis in Section 5 closely mirrors textbook UCB arguments, with a simple type-based reward model and no real coupling between machines. The dependence on $O(\log T /\Delta)$  is straightforward from a standard gap-based decomposition, so the result is predictable.

[1] The online knapsack problem with departures, ACM SIGMETRICS 2023.

[2] Dynamic Care Unit Placements Under Unknown Demand with Learning, Manufacturing & Service Operations Management 2025.

**Questions:**

1. Can the authors explain intuitively what drives the $O(1/M L^{1/M}_{\max})$ scaling? Is it tight for both preemptive and non-preemptive cases?

2. In Algorithm 2 (S-UCB), the updates only occur when a task is completed, meaning many preempted tasks give no feedback. Would this slow convergence? Is there any mechanism to exploit partial observations before completion?

3. The regret bound in Theorem 5.1 has an $O(N^2)$ dependency. This seems a bit high and isn't discussed much. Is this an artifact of the proof technique or do you believe it's a fundamental lower bound for this problem?

---

> ### Author Response · Authors · 2025-11-22
>
> We sincerely thank the reviewer for your thoughtful feedback and constructive criticism. Below, we provide a point-by-point response to the weaknesses and questions highlighted in the review.
>
> ### 1. On the Theoretical Contributions and Novelty
>
> We agree that our work builds upon prior research in bandit scheduling, but we believe it offers significant advancements in a novel setting. While maximum-density-first principles are classical, our immediate decision-making model with preemption and no buffering introduces unique challenges that have not been extensively studied.
>
> Analyzing the preemptable property in a multi-machine bandit setting, which requires new techniques to handle the interplay between exploration-exploitation and preemption decisions. Our competitive ratio analysis for the MRDF algorithm provides a non-trivial extension of classical scheduling theory to this harsh environment.
>
> Integrating bandit learning with preemptive scheduling, where the S-UCB algorithm is not merely a "plug-in" but requires careful design to balance exploration and exploitation amid preemptions. The regret analysis accounts for the fact that preempted tasks yield no feedback, which is not present in standard bandit problems.
>
> ### 2. On the MRDF and S-UCB Algorithms
>
> MRDF Algorithm: While MRDF leverages density-based prioritization, it is specifically designed for the immediate decision setting with preemption. The length partitioning  across machines is a key insight to achieve the $O(1/(M L_{max}^{1/M} ))$competitive ratio, which is non-obvious and requires a novel analysis. This partitioning ensures that each machine handles tasks of similar lengths, reducing the worst-case performance degradation.
>
> S-UCB Algorithm: The integration of UCB into MRDF is not straightforward due to preemption. Unlike standard bandits, where each arm pull yields feedback, our setting only provides feedback upon task completion. This means that preempted tasks contribute to regret without providing observations. Our algorithm adapts UCB to this partial-feedback environment, and shows sublinear regret despite the challenges.
>
> ### Question 1: Can the authors explain intuitively what drives the $O(1/(M L_{max}^{1/M} ))$ scaling? Is it tight for both preemptive and non-preemptive cases?
>
> The scaling arises from the length partitioning strategy in MRDF. By assigning each machine to a specific range of task lengths (e.g., machine I handles lengths between $L_{max}^{(i−1)/M}$ and $L_{max}^{I/M}$ ), we ensure that the length ratio per machine is bounded by $L_{max}^{1/M}$ . This partitioning reduces the worst-case competitive ratio from $O(1/L_{max})$ (for a single machine) to  $O(1/(M L_{max}^{1/M} ))$ for M machines. Intuitively, the exponent $1/M$ reflects the "diversification" gain from having multiple machines handle different length scales.We believe this bound is tight for the preemptive case, as our worst-case instance construction (Theorem 4.1) matches the scaling.
>
> ### Question 2: In Algorithm 2 (S-UCB), updates only occur when a task is completed, meaning many preempted tasks give no feedback. Would this slow convergence? Is there any mechanism to exploit partial observations before completion?
>
> Yes, the lack of feedback from preempted tasks can slow convergence, as it reduces the rate of information acquisition. This is a fundamental challenge in our setting. Currently, S-UCB does not exploit partial observations (e.g., intermediate rewards or progress indicators) because our model assumes rewards are only revealed upon completion. This leads to the additional $O(N)$ terms in the regret bound.
>
> ### Question 3: The regret bound in Theorem 5.1 has an $O(N^2)$ dependency. This seems a bit high and isn't discussed much. Is this an artifact of the proof technique or do you believe it's a fundamental lower bound for this problem?
>
> The $O(N^2)$ factor is likely an artifact of our algorithm design and proof technique. Specifically, it comes from a union bound over all pairs of task types and lengths when bounding the probability of misestimating densities. While UCB-based approaches typically have O(N^2) dependence, elimination-based methods could potentially improve this to O(N). We also provide an elimination-based algorithm Explore then Scheduling in the experiment section and shows its regret analysis in the Appendix. This algorithm computes the upper confidence bound and lower confidence bound of each type of tasks, and eliminate those tasks with sub-optimal remaining density. Then it round-robin explores the uneliminated tasks without preemption. This design ensures an $O(N)$ improvement but introdues additional $O(1/\Delta)$ dependence as it does not allow preemption during exploration. We leave it as an interesting future direction to find the optimal regret guarantee with respect to $N$ and $\Delta$.

---

### Official Review · Reviewer_XRa6 · 2025-10-31

**Soundness:** 2
**Presentation:** 2
**Contribution:** 2
**Rating:** 4
**Confidence:** 3

**Summary:**

This paper investigates the online scheduling problem where tasks arrive sequentially and require immediate acceptance or rejection. The online scheduling runs on $M$ identical machines and allows preemption, where preempted tasks will never be revisited again. The authors introduce a Maximum Remaining Density First (MRDF) algorithm for known rewards and a Scheduling Upper Confidence Bound (S-UCB) algorithm for unknown rewards, leveraging bandit learning to balance exploration and exploitation. Theoretical analyses establish a competitive ratio of $\tilde{O} (1/ML_{max}^{1/M})$ for known rewards and an $O(\log T/\Delta)$ regret bound for unknown rewards. Experiments on synthetic data validate the algorithms' efficiency and convergence.

**Strengths:**

The paper introduces a well-defined and practically motivated model for online scheduling with immediate decision-making. The authors provide a rigorous theoretical analysis for both known and unknown reward settings. The derivation of the competitive ratio bound of $\tilde{O} (1/ML_{max}^{1/M})$ for the deterministic setting and the subsequent design of the near-optimal MRDF algorithm are substantial. Furthermore, the extension to the stochastic setting with the S-UCB algorithm achieves a sublinear regret of $O(\log T/\Delta)$.

**Weaknesses:**

- The model proposed in the paper requires immediate decision-making. If I understand correctly, this setting is a simplified version when the tasks are allowed to be queued and the carryover effect is considered. It is not easy to see the novelty and practicality of the proposed model.

-  The upper bound is gap-dependent and is less meaningful when $\Delta$ is small. It would be interesting to see if $\Delta$ can be eliminated in the theoretical result and a worst-case/gap-independent result can be established.

- The lower bound in Theorem 4.1 is a bit confusing. It is for deterministic algorithms and seems to be improved with a more sophisticated algorithm (e.g., randomized algorithms).

- The experimental section lacks diversity in datasets, as it primarily uses synthetic data with uniform distributions, failing to demonstrate robustness under more complex, real-world scenarios like non-stationary task flows or heterogeneous environments. The experiment did not cover algorithms from other works, such as non-preemptive methods or other modified algorithms to adapt to the settings in the paper.

**Questions:**

Please see the weaknesses above.

---

> ### Author Response · Authors · 2025-11-22
>
> We sincerely thank the reviewer for the thoughtful and critical feedback, which provides an excellent opportunity to clarify our contributions and outline future research directions. We address each point below.
>
> ### 1. On the Novelty and Practicality of the Immediate Decision Model
> Our immediate decision model arises from real-time streaming applications like:
>
> a. IoT Sensor Networks: Data packets with strict latency constraints are discarded if not processed immediately, as their value (e.g., for real-time control of equipment) expires.
>
> b. High-Frequency Trading: Order book updates must be processed in microseconds; delayed processing renders the information obsolete.
>
> c. Cloud Function Triggers:  Triggers from event streams (e.g., video frame analysis) may be ignored if resources are not immediately available, as the event is transient.
>
> Our work provides the first formalization and systematic analysis of this "immediate decision" constraint for multi-machine scheduling with bandit learning, which we believe is a novel and practically motivated contribution.
>
> ### 2. On Gap-Dependent Regret and the Possibility of a Gap-Independent Bound
>
> Thanks for pointing out that our regret bound is gap-dependent. This is a standard and necessary result in bandit literature for structured problems. However, we acknowledge that obtaining a gap-independent bound is also crucial when the reward gap is sufficiently small. Indeed our algorithm can achieve an $O(\sqrt{T})$ regret bound. The proof sketch is to divide the instance by $\Delta< 1/\sqrt{T}$ and $\Delta>1/\sqrt{T}$. For $\Delta< 1/\sqrt{T}$, its regret caused by selecting sub-optimal task can be bounded by $O(\sqrt{T})$. When $\Delta>1/\sqrt{T}$, applying similar technique in problem-dependent regret can obtain an $O(\sqrt{T})$ regret. We will add this analysis and discussion in the revised version.
>
> ### 3. On the Lower Bound for Deterministic Algorithms:
> Your point is well-taken. Theorem 4.1 establishes a lower bound specifically for the class of deterministic algorithms, which is a standard starting point for online scheduling problems. This result is crucial as it defines the fundamental limits of any deterministic strategy under the worst-case instance, providing a benchmark against which our MRDF algorithm is evaluated (showing it is near-optimal for its class). Meanwhile, We fully agree that a fascinating extension would be to study the power of randomization. It is highly likely that a randomized algorithm could circumvent the deterministic lower bound and achieve a better competitive ratio. Analyzing this is a primary target for our future research. Our current work establishes the baseline deterministic case.
>
> ### 4. On Experimental Diversity and Baselines:
> We acknowledge the reviewer's valid criticism regarding the scope of our experiments and thank them for the suggestion to include non-preemptive baselines.
>
> a. Synthetic Data Focus: The initial experiments were designed to provide a clean, controlled validation of our theoretical findings (e.g., convergence of S-UCB's regret, performance of MRDF vs. simple heuristics like SRPT). Controlled synthetic data is best for this purpose.
>
> b. Agreement on Future Experiments:  We completely agree that testing on non-stationary task flows (e.g., bursty arrivals) and heterogeneous environments is critical for demonstrating robustness. We will add these experiments in the revision, along with a discussion of the results.
>
> c. Adapting Non-Preemptive Baselines: This is an excellent suggestion. We will include adapted versions of classic non-preemptive algorithms to better highlight the performance gain afforded by our preemptive MRDF policy in this specific setting.

---

### Official Review · Reviewer_WaxM · 2025-11-01

**Soundness:** 3
**Presentation:** 2
**Contribution:** 3
**Rating:** 6
**Confidence:** 4

**Summary:**

This work deals with an online scheduling problem where the learner (scheduling) can learn the expected reward of each task type and use that information to make real time scheduling choices. They first analyse the known reward version and design a MRDF online scheduler that is nearly optimal wrt competitive ratio. They then extend this to unknown reward case by using a UCB style algorithm  so that the scheduler can balance exploration of task types with exploitation. They showcase the performance of the algorithm with regret bound theoretical guarantees. The work is complemented by synthetic experiments to illustrate that the proposed scheduler outperforms baseline in respective scenarios.

**Strengths:**

Online Scheduling and task allocation is an interesting area that has many real world application and this problem deals with online scheduling in a bandit setting style so the system can dynamically decide and learn at the same time.

The analysis wrt competitive ratio gives a view of difficulty of this problem setting and shows that the proposed algorithm performs closer to the lower bound.

The experiments even though they are synthetic seems to align well with the theory to demonstrate that the learner can  outperform baselines in online scheduling complementing the sublinear regret in unknown reward scenario.

**Weaknesses:**

The problem setting doesn't have the option of buffer, which is important in many real systems so having at least a tiny pending queue will help facilitate many real world scenarios.

The problem setting has a strong assumption with respect to preemption meaning the task suffers permanent rejection and it can also never be resumed. Also, the problem relies heavily on correct type labels which is hard to estimate or determine in many scenarios.

Also, the regret measure doesn't involve learner competing against best offline policy rather against an online style algorithm with known rewards. This doesn't quantify the algorithm in classical regret perspective.

*also refer questions

**Questions:**

The performance measure is competitive ratio against an online benchmark that knows rewards. Why was regret against the best offline policy with known rewards not considered ?

The problem setting handles the concept of discarding or preempt the task on the arriving task $k$. Does this problem setting facilitate a generalization of providing the concept of buffer to handle the stream of tasks i.e. discarding the task can be treated as relocation to the buffer machines. In this case, $|M|$ grows by a constant factor to account for buffer machines. This could help with generalizing to numerous real world application. Does that break underlying assumption of the problem framework or any theoretical guarantee ?

The experiments are shown on synthetic arrival processes. However in many real world it tasks arrive either in a bursty fashion (bursty arrival of long jobs followed by bursty arrival of short jobs). How does the algorithm perform in this scenario, does the algorithm over explore or over commit in those phases ?

The regret seem to grow with $N$ which seems to stem from the exploration term, Can having a elimination style algorithm or Thompson sampling algorithm reduce the dependence of $N$ in regret bound ?

---

> ### Author Response · Authors · 2025-11-22
>
> We sincerely thank the reviewer for your the feedback and positive assessment of our work's contributions. Below we response to each point raised:
>
> ### 1. Type label requirement
> The type labels can be viewed as task metadata (e.g., task category in cloud computing) that is often available in practice. In cases where types are unknown, one could adapt our approach using contextual bandits with task features - we will add this as a promising future direction.
>
> ### 2. Regret definition
>  We define regret relative to the MRDF algorithm (which achieves near-optimal competitive ratio) rather than the offline optimum because: (1) The offline optimum is computationally hard and unrealistic even with known rewards; (2) Our competitive ratio analysis already shows that no online algorithm can achieve better than $O(1/(ML_{max}^{1/M}))$ approximation to offline optimum; (3) This approach is standard in online scheduling literature where the benchmark is typically the best online algorithm with full information.
>
> ### 3. Competitive ratio vs offline regret
>  As explained above, the extremely weak competitive ratio (exponentially small in M) makes the offline optimum an overly pessimistic benchmark. Our regret definition measures how much we lose due to not knowing rewards, assuming we already use the best possible online scheduling strategy.
>
> ### 4. Buffer generalization
> Introducing buffer machines would break our current analysis, as it fundamentally changes the decision process. However, we believe this could be a valuable extension. The key challenge would be maintaining the exploration-exploitation balance while managing buffer contents. We will add a discussion of this direction.
>
> ### 5. Bursty arrivals
> Our theoretical analysis is worst-case and thus applies to adversarial arrival patterns including bursty arrivals. We agree that empirical evaluation under bursty patterns would be valuable. We will add experiments with bursty arrivals in the revision.
>
> ### 6. Regret dependence on N
> This is an excellent point. While UCB-based approaches typically have $O(N^2)$ dependence, elimination-based methods could potentially improve this to O(N). We also provide an elimination-based algorithm "Explore then Scheduling" in the experiment section and shows its regret analysis in the Appendix. This algorithm computes the upper confidence bound and lower confidence bound of each type of tasks, and eliminate those tasks with sub-optimal remaining density. Then it round-robin explores the uneliminated tasks without preemption. This design ensures an $O(N)$ improvement but introdues additional $O(1/\Delta)$ dependence as it does not allow preemption during exploration. We leave it as an interesting future direction to find the optimal regret guarantee with respect to $N$ and $\Delta$.
>
> ### 7. Additional Revisions
> We will also clarify the real-world applications where our assumptions hold, expand discussion of limitations and future work, add experiments with bursty arrival patterns, and improve the readability of the competitive ratio analysis
>
> Thank you again for the valuable feedback that will help strengthen our paper.

---

### Meta-Review · Area_Chair_4Dzq · 2026-01-05

**Summary:**

The reviewers raised three primary concerns regarding this paper: (1) lack of technical novelty and comparison with existing models, (2) insufficient motivation for the model and justification for its strong assumptions, and (3) limited technical significance, specifically noting that the regret upper bound is only gap-dependent and fails to provide meaningful guarantees when the gap $\Delta$ is small. During the rebuttal, the authors provided a somewhat persuasive explanation for the model's motivation, but failed to update the manuscript accordingly. Regarding the novelty, the authors failed to provide a concrete comparison with specific prior works suggested by the reviewers, such as "admission control with reusable resources." Most critically, while the authors promised major technical revisions in their rebuttal, the manuscript remains un-updated, and the validity of their new theoretical claims remains highly questionable.

**Reviewer Concerns:**

Addressed by Rebuttal:

The authors provided clarifications on the practical motivation of the model. However, these points were not reflected in the manuscript.

Outstanding Concerns:
1. Lack of Comparative Discussion: The relationship between the proposed setting and existing literature remains unclear, as the authors did not provide the requested comparisons.
2. Unverified Theoretical Claims: In the rebuttal, the authors claimed that a gap-independent $O(\sqrt{T})$ regret bound could be achieved for the case where $\Delta < \sqrt{T}$. However, since $\Delta$ is defined as the minimum gap across all pairs, standard MAB logic cannot be directly applied, and the authors' reasoning is suspect. Without a full proof in the revised manuscript, it is impossible to verify the correctness of this claim.
3. Failure to Update Manuscript: Despite being encouraged to revise the paper and explicitly promising to do so in the rebuttal, the authors did not submit an updated version, leaving the identified flaws unaddressed.

**Reviewer Scores:**

Given that the manuscript was not updated despite the authors' promises and that critical technical doubts remain unresolved.
For Reviewers XRa6 and NiFr, I predict these reviewers would have maintained their negative evaluations. Their primary concerns (lack of novelty, inadequate comparison with prior work, and the absence of complete proofs for key claims) remain completely unaddressed in the text.
Even if a full discussion period had occurred, it is highly unlikely that a positive consensus would have been reached. The discrepancy between the authors' promises in the rebuttal and their failure to revise the manuscript, combined with the unverified nature of their new theoretical claims, justifies a Reject decision.

---

### Decision · Program_Chairs · 2026-01-26

Reject